# SHOULD ENSEMBLE MEMBERS BE CALIBRATED?

## ABSTRACT

Underlying the use of statistical approaches for a wide range of applications is the assumption that the probabilities obtained from a statistical model are representative of the "true" probability that event, or outcome, will occur. Unfortunately, for modern deep neural networks this is not the case, they are often observed to be poorly calibrated. Additionally, these deep learning approaches make use of large numbers of model parameters, motivating the use of Bayesian, or ensemble approximation, approaches to handle issues with parameter estimation. This paper explores the application of calibration schemes to deep ensembles from both a theoretical perspective and empirically on a standard image classification task, CIFAR-100. The underlying theoretical requirements for calibration, and associated calibration criteria, are first described. It is shown that well calibrated ensemble members will not necessarily yield a well calibrated ensemble prediction, and if the ensemble prediction is well calibrated its performance cannot exceed that of the average performance of the calibrated ensemble members. On CIFAR-100 the impact of calibration for ensemble prediction, and associated calibration is evaluated. Additionally the situation where multiple different topologies are combined together is discussed.

## 1 INTRODUCTION

Deep learning approaches achieve state-of-the-art performance in a wide range of applications, including image classification. However, these networks tend to be overconfident in their predictions, they often exhibit poor calibration. A system is well calibrated, if when the system makes a prediction with probability of 0.6 then 60% of the time that prediction is correct. Calibration is very important in deploying system, especially in risk-sensitive tasks, such as medicine (Jiang et al., 2012), auto-driving (Bojarski et al., 2016), and economics (Gneiting et al., 2007). It was shown by Niculescu-Mizil & Caruana (2005) that shallow neural networks are well calibrated. However, Guo et al. (2017) found that more complex neural network model with deep structures do not exhibit the same behaviour. This work motivated recent research into calibration for general deep learning systems. Previous research has mainly examined calibration based on samples from the true data distribution $\{\boldsymbol{x}^{(i)}, y^{(i)}\}_{i=1}^{N} \sim \mathrm{p}(\boldsymbol{x}, \boldsymbol{\omega}), y^{(i)} \in \{\omega_1, ..., \omega_K\}$ (Zadrozny & Elkan, 2002; Vaicenavicius et al., 2019). This analysis relies on the limiting behaviour as $N \rightarrow +\infty$ to define a well calibrated system

$$\mathrm{P}(y = \hat{y} | \mathrm{P}(\hat{y}|\boldsymbol{x}; \boldsymbol{\theta}) = p) = p \iff \lim_{N \to +\infty} \sum_{i \in \mathcal{S}_j^p} \frac{\delta(y^{(i)}, \hat{y}^{(i)})}{|\mathcal{S}_j^p|} = p \qquad (1)$$

where $\mathcal{S}_j^p = \{i | \mathrm{P}(\hat{y}^{(i)} = j | \boldsymbol{x}^{(i)}; \boldsymbol{\theta}) = p, i = 1, ..., N\}$ and $\hat{y}^{(i)}$ the model prediction for $\boldsymbol{x}^{(i)}$. $\delta(s, t) = 1$ if $s = t$, otherwise 0. However, Eq. (1) doesn't explicitly reflect the relation between $\mathrm{P}(y = \hat{y} | \mathrm{P}(\hat{y}|\boldsymbol{x}; \boldsymbol{\theta}) = p)$ and the underlying data distribution $\mathrm{p}(\boldsymbol{x}, y)$. In this work we examine this explicit relationship and use it to define a range of calibration evaluation criteria, including the standard sample-based criteria.

One issue with deep-learning approaches is the large number of model parameters associated with the networks. Deep ensembles (Lakshminarayanan et al., 2017) is a simple, effective, approach for handling this problem. It has been found to improve performance, as well as allowing measures of uncertainty. In recent literature there has been "contradictory" empirical observations about the relationship between the calibration of the members of the ensemble and the calibration of the final

ensemble prediction (Rahaman & Thiery, 2020; Wen et al., 2020). In this paper, we examine the underlying theory and empirical results relating to calibration with ensemble methods. We found, both theoretically and empirically, that ensembling multiple calibrated models decreases the confidence of final prediction, resulting in an ill-calibrated ensemble prediction. To address this, strategies to calibrate the final ensemble prediction, rather than individual members, are required. Additionally we empiricaly examine the situation where the ensemble is comprised of models with different topologies, and resulting complexity/performance, requiring non-uniform ensemble averaging.

In this study, we focus on post-hoc calibration of ensemble, based on temperature annealing. Guo et al. (2017) conducted a thorough comparison of various existing post-hoc calibration methods and found that temperature scaling was a simple, fast, and often highly effective approach to calibration. However, standard temperature scaling acts globally for all regions of the input samples, i.e. all logits are scaled towards one single direction, either increasing or decreasing the distribution entropy. To address this constraint, that may hurt some legitimately confident predictions, we investigate the effect of region-specific temperatures. Empirical results demonstrate the effectiveness of this approach, with minimal increase in the number of calibration parameters.

## 2 RELATED WORK

Calibration is inherently related to uncertainty modeling. Two of the most important scopes of calibration are calibration evaluation and calibration system construction. One method to assessing calibration is the reliability diagram (Vaicenavicius et al., 2019; Bröcker, 2012). Though informative, It is still desirable to have an overall metric. Widmann et al. (2019) investigate different distances in the probability simplex for estimating calibration error. Nixon et al. (2019) point out the problem of fixed spaced binning scheme, bins with few predictions may have low-bias but high-variance measurement. Calibration error measure adaptive to dense populated regions have also been proposed (Nixon et al., 2019). Vaicenavicius et al. (2019) treated the calibration evaluation as hypotheses tests. All these approaches examine calibration criteria from a sample-based perspective, rather than as a function of the underlying data distribution which is used in the thoretical analysis in this work.

There are two main approaches to calibrating systems. The first is to recalibrate the uncalibrated systems with post-hoc calibration mapping, e.g. Platt scaling (Platt et al., 1999), isotonic regression (Zadrozny & Elkan, 2002), Dirichlet calibration (Kull et al., 2017; 2019). The second is to directly build calibrated systems, via: (i) improving model structures, e.g. deep convolutional Gaussian processes (Tran et al., 2019); (ii) data augmentation, e.g. adversarial samples (Hendrycks & Dietterich, 2019; Stutz et al., 2020) or Mixup (Zhang et al., 2018); (iii) minimize calibration error during training (Kumar et al., 2018). Calibration based on histogram binning (Zadrozny & Elkan, 2001), Bayesian binning (Naeini et al., 2015) and scaling binning (Kumar et al., 2019) are related to our proposed dynamic temperature scaling, in the sense that the samples are divided into regions and separate calibration mapping are applied. However, our method can preserve the property that all predictions belonging to one sample sum to 1. The region-based classifier by Kuleshov & Liang (2015) is also related to our approach.

Ensemble diversity has been proposed for improved calibration (Raftery et al., 2005; Stickland & Murray, 2020). In Zhong & Kwok (2013), ensembles of SVM, logistic regressor, boosted decision trees are investigated, where the combination weights of calibrated probabilities is based on AUC of ROC. However, AUC is not comparable between different models as discussed in Ashukha et al. (2020). In this work we investigate the combination of different deep neural network structures. The weights assigned to the probabilities is optimised using a likelihood-based metric.

## 3 CALIBRATION FRAMEWORK

Let $\mathcal{X} \subseteq \mathbb{R}^d$ be the $d$-dimensional input space and $\mathcal{Y} = \{\omega_1, ..., \omega_K\}$ be the discrete output space consisting of $K$ classes. The true underlying joint distribution for the data is $\mathrm{p}(\boldsymbol{x}, \omega) = \mathrm{P}(\omega|\boldsymbol{x})\mathrm{p}(\boldsymbol{x}), \boldsymbol{x} \in \mathcal{X}, \omega \in \mathcal{Y}$. Given some training data $\mathcal{D} \sim \mathrm{p}(\boldsymbol{x}, \omega)$, a model $\boldsymbol{\theta}$ is trained to predict the distribution $\mathrm{P}(\boldsymbol{\omega}|\boldsymbol{x}; \boldsymbol{\theta})$ given observation features. For a calibrated system the average predicted posterior probability should equate to the average posterior of the underlying distribution for a specific probability region. Two extreme cases will always yield perfect calibration. First when the predictions that are the same, and equal to the class prior for all inputs, $\mathrm{P}(\omega_j|\boldsymbol{x}; \boldsymbol{\theta}) = \mathrm{P}(\omega_j)$. Sec-

ond the minimum Bayes' risk classifier is obtained, $P(\omega_j|\boldsymbol{x};\boldsymbol{\theta}) = \frac{p(\boldsymbol{x},\omega_j)}{\sum_{k=1}^{K} p(\boldsymbol{x},\omega_k)}$. Note that perfect calibration doesn't imply high accuracy, as shown by the system predicting the prior distribution.

### 3.1 DISTRIBUTION CALIBRATION

A system is calibrated if the predictive probability values can accurately indicate the portion of correct predictions. *Perfect calibration* for a system that yields $P(\boldsymbol{\omega}|\boldsymbol{x};\boldsymbol{\theta})$ when the training and test data are obtained form the joint distribution $p(\boldsymbol{x},\boldsymbol{\omega})$ can be defined as:

$$\int_{\boldsymbol{x}\in\mathcal{R}_j^p(\boldsymbol{\theta},\epsilon)} P(\omega_j|\boldsymbol{x};\boldsymbol{\theta})p(\boldsymbol{x})\mathrm{d}\boldsymbol{x} = \int_{\boldsymbol{x}\in\mathcal{R}_j^p(\boldsymbol{\theta},\epsilon)} P(\omega_j|\boldsymbol{x})p(\boldsymbol{x})\mathrm{d}\boldsymbol{x} \quad \forall p, \omega_j, \epsilon \to 0 \tag{2}$$

$$\mathcal{R}_j^p(\boldsymbol{\theta},\epsilon) = \left\{\boldsymbol{x} \Big| |P(\omega_j|\boldsymbol{x};\boldsymbol{\theta}) - p| \le \epsilon, \boldsymbol{x} \in \mathcal{X}\right\} \tag{3}$$

$\mathcal{R}_j^p(\boldsymbol{\theta},\epsilon)$ denotes the region of input space where the system predictive probability for class $\omega_j$ is sufficiently close, within error of $\epsilon$, to the probability $p$. A perfectly calibrated system will satisfy this expression for all regions, the expected predictive probability (left side of Eq. (2)) is identical to the expected correctness, i.e., expected true probability (right side of Eq. (2)).

$\mathcal{R}_j^p(\boldsymbol{\theta},\epsilon)$ defines the region in which calibration is defined. For *top-label calibration*, only the most probable class is considered and the region defined in Eq. (3) is modified to reflect this:

$$\tilde{\mathcal{R}}_j^p(\boldsymbol{\theta},\epsilon) = \mathcal{R}_j^p(\boldsymbol{\theta},\epsilon) \cap \left\{\boldsymbol{x}\Big|\omega_j = \arg\max_{\omega} P(\omega|\boldsymbol{x};\boldsymbol{\theta}), \boldsymbol{x}\in\mathcal{X}\right\} \tag{4}$$

Eq. (4) is a strict subset of Eq. (3). As the two calibration regions are different between calibration and top-label calibration, perfect calibration doesn't imply top-label calibration, and vise versa. A simple illustrative example of this property is given in A.3. Binary classification, $K = 2$, is an exception to this general rule, as the regions for top-label calibration are equivalent to those for perfect calibration, i.e. $\tilde{\mathcal{R}}_j^p(\boldsymbol{\theta},\epsilon) = \mathcal{R}_j^p(\boldsymbol{\theta},\epsilon)$. Hence, perfect calibration is equivalent to top-label calibration for binary classification (Nguyen & O'Connor, 2015).

Eq. (2) defines the requirements for a perfectly calibrated system. It is useful to define metrics that allow how close a system is to perfect calibration to be assessed. Let the region calibration error be:

$$\mathcal{C}_j^p(\boldsymbol{\theta},\epsilon) = \int_{\boldsymbol{x}\in\mathcal{R}_j^p(\boldsymbol{\theta},\epsilon)} (P(\omega_j|\boldsymbol{x};\boldsymbol{\theta}) - P(\omega_j|\boldsymbol{x}))p(\boldsymbol{x})\mathrm{d}\boldsymbol{x} \tag{5}$$

This then allows two forms of expected calibration losses to be defined

$$\texttt{ACE}(\boldsymbol{\theta}) = \frac{1}{K}\int_0^1 \left|\sum_{j=1}^{K}\mathcal{C}_j^p(\boldsymbol{\theta},\epsilon)\right|\mathrm{d}p; \quad \texttt{ACCE}(\boldsymbol{\theta}) = \frac{1}{K}\sum_{j=1}^{K}\int_0^1 \left|\mathcal{C}_j^p(\boldsymbol{\theta},\epsilon)\right|\mathrm{d}p \tag{6}$$

All Calibration Error (ACE) only considers the expected calibration error for a particular probability, irrespective of the class associated with the data[1] (Hendrycks et al., 2019). Hence, All Class Calibration Error (ACCE) that requires that all classes minimises the calibration error for all probabilities is advocated by Kull et al. (2019); Kumar et al. (2019). Nixon et al. (2019) propose the Thresholded Adaptive Calibration Error (TACE) to consider only the prediction larger than a threshold, and it can be described as a special case of ACCE by replacing the integral range. Naeini et al. (2015) also propose to only consider the region with maximum error.

Though measures such as ACE and ACCE require consistency of the expected posteriors with the true distribution, for tasks with multiple classes, particularly large numbers of classes, the same weight is given to the ability of the model to assign low probabilities to highly unlikely classes, and high probabilities to the "correct" class. For systems with large numbers of classes this can yield artificially low scores. To address this problem it is more common to replace the regions in Eq. (5) with the top-label regions in Eq. (4), to give a top-label calibration error $\tilde{\mathcal{C}}_j^p(\boldsymbol{\theta},\epsilon)$. This then yields

---

[1]In this section the references given refer to the sample-based equivalent versions of the distributional calibration expressions in this paper using the same concepts, rather than identical expressions.

the expected top-label equivalents of ACCE and ACE, Expected Class Calibration Error (ECCE) and Expected Calibration Error (ECE). Here for example ECE by Guo et al. (2017) is expressed as

$$\text{ECE}(\boldsymbol{\theta}) = \int_0^1 \left| \sum_{j=1}^K \int_{\boldsymbol{x} \in \tilde{\mathcal{R}}_j^p(\boldsymbol{\theta}, \epsilon)} (\text{P}(\omega_j | \boldsymbol{x}; \boldsymbol{\theta}) - \text{P}(\omega_j | \boldsymbol{x})) \text{p}(\boldsymbol{x}) \text{d}\boldsymbol{x} \right| \text{d}p \tag{7}$$

$$= \int_0^1 \mathcal{O}(\boldsymbol{\theta}, p) |\text{Conf}(\boldsymbol{\theta}, p) - \text{Acc}(\boldsymbol{\theta}, p)| \text{d}p \tag{8}$$

where $\mathcal{O}(\boldsymbol{\theta}, p) = \sum_{j=1}^K \int_{\boldsymbol{x} \in \tilde{\mathcal{R}}_j^p(\boldsymbol{\theta}, \epsilon)} \text{p}(\boldsymbol{x}) \text{d}\boldsymbol{x}$ is the fraction observations that are assigned to that particular probability and $\text{Conf}(\boldsymbol{\theta}, p)$ and $\text{Acc}(\boldsymbol{\theta}, p)$ are the ideal distribution accuracy and confidences from the model for that probability. For more details see the appendix.

## 3.2 SAMPLE-BASED CALIBRATION

Usually only samples from the true joint distribution are available. Any particular training set is drawn from the distribution to yield

$$\mathcal{D} = \left\{ \{ \boldsymbol{x}^{(i)}, y^{(i)} \} \right\}_{i=1}^N, \quad \{ \boldsymbol{x}^{(i)}, y^{(i)} \} \sim \text{p}(\boldsymbol{x}, \boldsymbol{\omega}), \quad y^{(i)} \in \{ \omega_1, ..., \omega_K \}.$$

The region defined in Eq. (3) is now changed to be indices of the samples:

$$\mathcal{S}_j^p(\boldsymbol{\theta}, \epsilon) = \left\{ i \middle| |\text{P}(\omega_j | \boldsymbol{x}^{(i)}; \boldsymbol{\theta}) - p| \le \epsilon, \boldsymbol{x}^{(i)} \in \mathcal{D} \right\}, \tag{9}$$

The sample-based version of "perfect" calibration in Eq. (2) can then be expressed as:

$$\frac{1}{|\mathcal{S}_j^p(\boldsymbol{\theta}, \epsilon)|} \sum_{i \in \mathcal{S}_j^p(\boldsymbol{\theta}, \epsilon)} \text{P}(\omega_j | \boldsymbol{x}^{(i)}; \boldsymbol{\theta}) = \frac{1}{|\mathcal{S}_j^p(\boldsymbol{\theta}, \epsilon)|} \sum_{i \in \mathcal{S}_j^p(\boldsymbol{\theta}, \epsilon)} \delta(y^{(i)}, \omega_j), \quad \forall p, \omega_j, \epsilon \to 0 \tag{10}$$

as $N \to \infty$. When considering finite data, in this case $N$ samples, it is important to set $\epsilon$ appropriately. Setting different $\epsilon$ yields different regions and leads to different calibration results (Kumar et al., 2019). Thus it is important to specify $\epsilon$ when defining calibration for a system.

Similarly, the distribution form of top-label calibration can be written in terms of samples as Eq. (4), with different regions considered:

$$\tilde{\mathcal{S}}_j^p(\boldsymbol{\theta}, \epsilon) = \mathcal{S}_j^p(\boldsymbol{\theta}, \epsilon) \cap \left\{ i \middle| \omega_j = \arg\max_\omega \text{P}(\omega | \boldsymbol{x}^{(i)}; \boldsymbol{\theta}), \boldsymbol{x}^{(i)} \in \mathcal{D} \right\} \tag{11}$$

The sample-based calibration losses in region $\mathcal{S}_j^p(\boldsymbol{\theta}, \epsilon)$ can be defined based on Eq. (10). For example ACE in Eq. (6) can be expressed in its sample-based form (Hendrycks et al., 2019)

$$\text{ACE}(\boldsymbol{\theta}, \epsilon) = \frac{1}{NK} \sum_{p \in \mathcal{P}(\epsilon)} \left| \sum_{j=1}^K \sum_{i \in \mathcal{S}_j^p(\boldsymbol{\theta}, \epsilon)} \left( \text{P}(\omega_j | \boldsymbol{x}^{(i)}; \boldsymbol{\theta}) - \delta(y^{(i)}, \omega_j) \right) \right| \tag{12}$$

where $\mathcal{P}(\epsilon) = \{ p | p = \min\{1, (2z-1)\epsilon\}, z \in \mathbb{Z}^+ \}$, and $\mathbb{Z}^+$ is the set of positive integers. The measure of ECE relating to Eq. (7), which only considers the top regions in Eq. (11) can be defined as Guo et al. (2017)

$$\text{ECE}(\boldsymbol{\theta}, \epsilon) = \frac{1}{N} \sum_{p \in \mathcal{P}(\epsilon)} \left| \sum_{j=1}^K \sum_{i \in \tilde{\mathcal{S}}_j^p(\boldsymbol{\theta}, \epsilon)} \left( \text{P}(\omega_j | \boldsymbol{x}^{(i)}; \boldsymbol{\theta}) - \delta(y^{(i)}, \omega_j) \right) \right| \tag{13}$$

$$= \sum_{p \in \mathcal{P}(\epsilon)} \frac{\left( \sum_{j=1}^K |\tilde{\mathcal{S}}_j^p(\boldsymbol{\theta}, \epsilon)| \right)}{N} \left| \text{Conf}(\boldsymbol{\theta}, p) - \text{Acc}(\boldsymbol{\theta}, p) \right| \tag{14}$$

It should be noted that for a finite number of samples, the regions $\mathcal{S}_j^p(\boldsymbol{\theta}, \epsilon)$ and $\tilde{\mathcal{S}}_j^p(\boldsymbol{\theta}, \epsilon)$ derived from the samples can be different from the theoretical regions, leading to difference between theoretical calibration error measures and the values estimated from the finite samples. This is also referred to

as "estimator randomness" by Vaicenavicius et al. (2019). An example is given in A.3 to illustrate this mismatch.

The simplest region specification for calibration is to set $\epsilon = 1$. In this case, $|\mathcal{S}_j^p(\boldsymbol{\theta}, 1)| = N$, and the "minimum" perfect calibration requirement for a system with parameters $\boldsymbol{\theta}$ becomes

$$\frac{1}{N} \sum_{i=1}^{N} \mathrm{P}(\omega_j | \boldsymbol{x}^{(i)}; \boldsymbol{\theta}) = \frac{1}{N} \sum_{i=1}^{N} \delta(y^{(i)}, \omega_j), \quad \forall \omega_j \tag{15}$$

This is also referred to as *global calibration* in this paper. Similarly, *global top-label calibration* can be defined as

$$\frac{1}{N} \sum_{i=1}^{N} \mathrm{P}(\hat{y}^{(i)} | \boldsymbol{x}^{(i)}; \boldsymbol{\theta}) = \frac{1}{N} \sum_{i=1}^{N} \delta(y^{(i)}, \hat{y}^{(i)}), \quad \hat{y}^{(i)} = \arg \max_{\omega} \mathrm{P}(\omega | \boldsymbol{x}^{(i)}; \boldsymbol{\theta}) \tag{16}$$

## 4 ENSEMBLE CALIBRATION

An interesting question when using ensembles is whether calibrating the ensemble members is sufficient to ensure calibrated predictions. Initially the ensemble model will be viewed as an approximation to Bayesian parameter estimation. Given training data $\mathcal{D}$, the prediction of class $\omega_j$ is:

$$\begin{aligned}
\mathrm{P}(\omega_j | \boldsymbol{x}^*, \mathcal{D}) &= \mathbb{E}_{\boldsymbol{\theta} \sim \mathrm{p}(\boldsymbol{\theta}|\mathcal{D})}[\mathrm{P}(\omega_j | \boldsymbol{x}^*; \boldsymbol{\theta})] = \int \mathrm{P}(\omega_j | \boldsymbol{x}^*; \boldsymbol{\theta}) \mathrm{p}(\boldsymbol{\theta}|\mathcal{D}) \mathrm{d}\boldsymbol{x} \\
&\approx P(\omega_j | \boldsymbol{x}^*; \boldsymbol{\Theta}) = \frac{1}{M} \sum_{m=1}^{M} \mathrm{P}(\omega_j | \boldsymbol{x}^*; \boldsymbol{\theta}^{(m)}); \quad \boldsymbol{\theta}^{(m)} \sim \mathrm{p}(\boldsymbol{\theta}|\mathcal{D})
\end{aligned} \tag{17}$$

where Eq. (17) is an ensemble, Monte-Carlo, approximation to the full Bayesian integration, with $\boldsymbol{\theta}^{(m)}$ the $m$-th ensemble member parameters in the ensemble $\boldsymbol{\Theta}$. The predictions of ensemble and members are $\hat{y}_m^* = \arg \max_{\omega} \{\mathrm{P}(\omega | \boldsymbol{x}^*; \boldsymbol{\theta}^{(m)})\}, \hat{y}_{\mathrm{E}}^* = \arg \max_{\omega} \left\{ \frac{1}{M} \sum_{m=1}^{M} \mathrm{P}(\omega | \boldsymbol{x}^*; \boldsymbol{\theta}^{(m)}) \right\}$.

### 4.1 THEORETICAL ANALYSIS

For ensemble methods it is only important that the final ensemble prediction, $\hat{y}_{\mathrm{E}}$, is well calibrated, rather than the individual ensemble members. It is useful to examine the relationship between this ensemble prediction and the predictions from the individual models when the ensemble members are calibrated. Consider a particular top-label calibration region for the ensemble prediction, $\tilde{\mathcal{R}}^p(\boldsymbol{\Theta}, \epsilon)$, related to Eq. (4), the following expression is true

$$\int_{\boldsymbol{x} \in \tilde{\mathcal{R}}^p(\boldsymbol{\Theta}, \epsilon)} \frac{1}{M} \sum_{m=1}^{M} \mathrm{P}(\hat{y}_{\mathrm{E}} | \boldsymbol{x}; \boldsymbol{\theta}^{(m)}) \mathrm{p}(\boldsymbol{x}) \mathrm{d}\boldsymbol{x} \leq \int_{\boldsymbol{x} \in \tilde{\mathcal{R}}^p(\boldsymbol{\Theta}, \epsilon)} \frac{1}{M} \sum_{m=1}^{M} \mathrm{P}(\hat{y}_m | \boldsymbol{x}; \boldsymbol{\theta}^{(m)}) \mathrm{p}(\boldsymbol{x}) \mathrm{d}\boldsymbol{x} \tag{18}$$

where the ensemble region is defined as $\tilde{\mathcal{R}}^p(\boldsymbol{\Theta}, \epsilon) = \left\{ \boldsymbol{x} \Big| |\mathrm{P}(\hat{y}_{\mathrm{E}} | \boldsymbol{x}; \boldsymbol{\Theta}) - p| \leq \epsilon, \boldsymbol{x} \in \mathcal{X} \right\}$. For all regions $\tilde{\mathcal{R}}^p(\boldsymbol{\Theta}, \epsilon)$ the ensemble is no more confident than the average confidence of individual member predictions. This puts bounds on the ensemble prediction performance if the resulting ensemble prediction is top-label calibrated, and all ensemble members yield the same region $\tilde{\mathcal{R}}^p(\boldsymbol{\Theta}, \epsilon)$. Here

$$\int_{\boldsymbol{x} \in \tilde{\mathcal{R}}^p(\boldsymbol{\Theta}, \epsilon)} \mathrm{P}(\hat{y}_{\mathrm{E}} | \boldsymbol{x}; \boldsymbol{\Theta}) \mathrm{p}(\boldsymbol{x}) \mathrm{d}\boldsymbol{x} = \int_{\boldsymbol{x} \in \tilde{\mathcal{R}}^p(\boldsymbol{\Theta}, \epsilon)} \mathrm{P}(\hat{y}_{\mathrm{E}} | \boldsymbol{x}) \mathrm{p}(\boldsymbol{x}) \mathrm{d}\boldsymbol{x} \tag{19}$$

From Eq. (18) the left hand-side of this expression, the ensemble prediction confidence, cannot be greater that than the average ensemble member confidence. If the regions associated with the ensemble prediction and members are the same, then for top-label calibrated members this average confidence is the same as the average ensemble member accuracy. Furthermore, if the ensemble prediction is top-label calibrated, then this average ensemble member accuracy bounds the ensemble prediction accuracy. Under these conditions ensembling the members yields no performance gains.

The above bound holds with the assumption that the members are calibrated on the same regions. Proposition 3 in Appendix describes one trivial case when all members are calibrated on the same regions. Another case is the calibration on global regions. As shown in Proposition 1, at the global level, ensemble accuracy is still bounded.

**Proposition 1.** If all members and the corresponding ensemble are globally top-label calibrated, the ensemble performance is no better than the average performance of the members:

$$\frac{1}{N}\sum_{i=1}^{N}\delta(y^{(i)}, \hat{y}_{\mathrm{E}}^{(i)}) \quad \leq \quad \frac{1}{M}\sum_{m=1}^{M}\left(\frac{1}{N}\sum_{i=1}^{N}\delta(y^{(i)}, \hat{y}_{m}^{(i)})\right) \tag{20}$$

*Proof.* If all members and the ensemble are globally top-label calibrated,

$$\frac{1}{N}\sum_{i=1}^{N}\mathrm{P}(\hat{y}_{m}^{(i)}|\boldsymbol{x}^{(i)};\boldsymbol{\theta}^{(m)}) \quad = \quad \frac{1}{N}\sum_{i=1}^{N}\delta(y^{(i)}, \hat{y}_{m}^{(i)}), \quad m = 1, ..., M \tag{21}$$

$$\frac{1}{N}\sum_{i=1}^{N}\left(\frac{1}{M}\sum_{m=1}^{M}\mathrm{P}(\hat{y}_{\mathrm{E}}^{(i)}|\boldsymbol{x}^{(i)};\boldsymbol{\theta}^{(m)})\right) \quad = \quad \frac{1}{N}\sum_{i=1}^{N}\delta(y^{(i)}, \hat{y}_{\mathrm{E}}^{(i)}) \tag{22}$$

By definition,

$$\mathrm{P}(\hat{y}_{\mathrm{E}}^{(i)}|\boldsymbol{x}^{(i)};\boldsymbol{\theta}^{(m)}) \leq \mathrm{P}(\hat{y}_{m}^{(i)}|\boldsymbol{x}^{(i)};\boldsymbol{\theta}^{(m)}) \tag{23}$$

Hence,

$$\frac{1}{N}\sum_{i=1}^{N}\delta(y^{(i)}, \hat{y}_{\mathrm{E}}^{(i)}) \quad \leq \quad \frac{1}{M}\sum_{m=1}^{M}\left(\frac{1}{N}\sum_{i=1}^{N}\delta(y^{(i)}, \hat{y}_{m}^{(i)})\right) \tag{24}$$

$\square$

However, this is not true for all-label calibration. In both cases, all-label calibrated members always yield all-label calibrated ensemble, no matter whether the ensemble accuracy exceeds the mean accuracy of members or not (Example 2 in Appendix gives illustration on a synthetic dataset).

**Proposition 2.** If all members are global all-label calibrated, then the overall ensemble is global all-label calibrated.

*Proof.* If all members are global all-label calibrated, then

$$\frac{1}{N}\sum_{i=1}^{N}\mathrm{P}(\omega_j|\boldsymbol{x}^{(i)};\boldsymbol{\theta}^{(m)}) \quad = \quad \frac{1}{N}\sum_{i=1}^{N}\delta(y^{(i)}, \omega_j), \quad \forall \omega_j, \ m = 1, ..., M \tag{25}$$

Hence,

$$\frac{1}{N}\sum_{i=1}^{N}\mathrm{P}(\omega_j|\boldsymbol{x}^{(i)};\boldsymbol{\Theta}) = \frac{1}{M}\sum_{m=1}^{M}\left(\frac{1}{N}\sum_{i=1}^{N}\delta(y^{(i)}, \omega_j)\right) = \frac{1}{N}\sum_{i=1}^{N}\delta(y^{(i)}, \omega_j) \tag{26}$$

$\square$

In general the regions are not the same, the ensemble accuracy is not bounded in the above way. However, note that global level calibration is the minimum requirement of calibration. The above discussion based on regions still sheds light on the question of should the members be calibrated or not, though the final theoretical answer is still absent. It should be also noted that, global all-label calibration does not imply global top-label calibration, because the regions considered are different (as illustrated by Example 1 in Appendix). For the discussion so far, the ensemble members are combined with uniform weights, motivated from a Bayesian approximation perspective. When, for example, multiple different topologies are used as members of the ensemble, a non-uniform averaging of the members of the ensemble, reflecting the model complexities and performance may be useful. Propositions 1 and 2 will still apply.

## 4.2 TEMPERATURE ANNEALING FOR ENSEMBLE CALIBRATION

Calibrating ensembles can be performing using a function $f \in \mathcal{F}$ with some parameters, $\boldsymbol{t}$, $\mathcal{F}$ : $[0,1] \rightarrow [0,1]$ for scaling probabilities. There are two modes for calibrating an ensemble:

**Pre-combination Mode.** the function is applied to the probabilities predicted by members, prior to combining the members to obtain ensemble prediction using a set of calibration parameters $\boldsymbol{T}$.

$$\mathrm{P}_{\mathrm{pre}}(\hat{y}_{\mathrm{E}}|\boldsymbol{x};\boldsymbol{\Theta},\boldsymbol{T}) = \frac{1}{M}\sum_{m=1}^{M} f\left(\mathrm{P}(\hat{y}_{\mathrm{E}}|\boldsymbol{x};\boldsymbol{\theta}^{(m)}), \boldsymbol{t}^{(m)}\right) \qquad (27)$$

**Post-combination Mode.** the function is applied to the ensemble predicted probability after combining members' predictions.

$$\mathrm{P}_{\mathrm{post}}(\hat{y}_{\mathrm{E}}|\boldsymbol{x};\boldsymbol{\Theta},\boldsymbol{t}) = f\left(\left(\frac{1}{M}\sum_{m=1}^{M}\mathrm{P}(\hat{y}_{\mathrm{E}}|\boldsymbol{x};\boldsymbol{\theta}^{(m)})\right),\boldsymbol{t}\right) \qquad (28)$$

There are many functions for transforming predicted probability in the calibration literature, e.g. histogram binning, Platt scaling and temperature annealing. However, histogram binning shouldn't be adopted in the pre-combination mode as scaling function $f$ for calibrating multi-class ensemble, as the transformed values may not yield a valid PMF.

As shown in Guo et al. (2017), temperature scaling is a simple, effective, option for the mapping function $\mathcal{F}$, which scales the logit values associated with the posterior by a temperature $t$, $f(\boldsymbol{z};t) = \exp\{\boldsymbol{z}/t\}/\sum_j \exp\{z_j/t\}$. Here a single temperature is used for scaling logits for all samples. This leads to the problem that the entropy of the predictions for all regions are either increased or decreased. From Eq. (2) the temperature can be made region specific.

$$f_{\mathrm{dyn}}(\boldsymbol{z};\boldsymbol{t}) = \frac{\exp\{\boldsymbol{z}/t_r\}}{\sum_j \exp\{z_j/t_r\}}, \quad \text{if } \max_i \frac{\exp\{z_i\}}{\sum_j \exp\{z_j\}} \in \mathcal{R}_r \qquad (29)$$

To determine the optimal set of temperatures, the samples in the validation set are divided into $R$ regions based on the ensemble predictions (e.g. $\mathcal{R}_1 = [0, 0.3)$, $\mathcal{R}_2 = [0.3, 0.6)$, and $\mathcal{R}_3 = [0.6, 1]$). Each region has an individual temperature for scaling $\{\mathcal{R}_r, t_r\}_{r=1}^{R}$.

## 4.3 EMPIRICAL RESULTS

Experiments were conducted on CIFAR-100 (and CIFAR-10 in the A.4). The data partition was 45,000/5,000/10,000 images for train/validation/test. We train LeNet (LEN) (LeCun et al., 1998), DenseNet 100 and 121 (DSN100, DSN121), (Huang et al., 2017) and Wide ResNet 28 (RSN) (Zagoruyko & Komodakis, 2016) following the original training recipes in each paper (more details in A.4). The results presented are slightly lower than that in the original papers, as 5,000 images were held-out to enable calibration parameter optimisation.

Figure 1 examines the empirical performance of ensemble calibration on CIFAR-100 test set using the three trained networks. The top row shows that, with appropriate temperature scaling, the members are calibrated on different regions (because otherwise the accuracy values should be the same). The middle row shows the ECE of ensemble members and ensemble prediction at different temperatures. The optimal calibration temperature for the ensemble prediction are consistently smaller than those associated with the ensemble members. This indicates that the ensemble predictions are less confident than those of the members, as stated in Eq. (23). The bottom row of figures show the reliability curves when the ensemble members are calibrated with optimal temperature values, and the resulting combination. It is clear that calibrating the ensemble members, using temperature, does not yield a calibrated ensemble prediction. Furthermore for all models the ensemble prediction is less confident than it should be, the line is above the diagonal. As discussed in Proposition 1, this is necessary, or the ensemble prediction is no better, which is clearly not the case for the performance plots in the top row. This ensemble performance is relatively robust to poorly calibrated ensemble members, with consistent performance over a wide range of temperatures.

Table 1 shows the calibration performance using three temperature scaling methods, pre-, post- and dynamic post-combination. The temperatures are optimized to minimize ECE (Liang et al.,

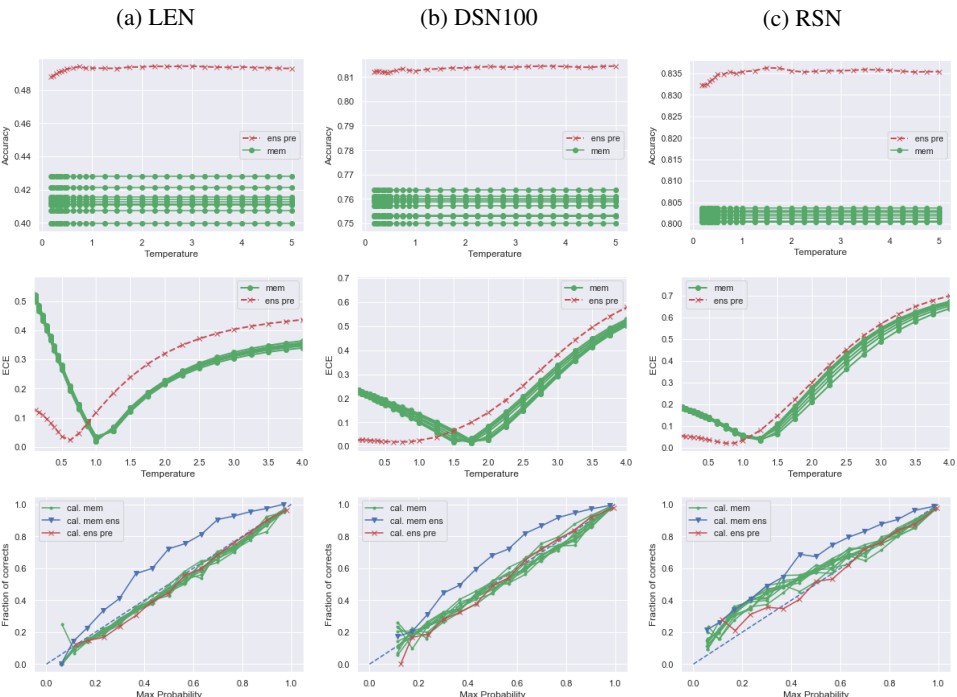

Figure 1: Top-label calibration error and accuracy of members (mem) and the whole ensemble (ens) on CIFAR-100 (test set) using LeNet, DenseNet and ResNet. "pre" denotes the calibration where shared temperature is applied to members before combination. The reliability curves shows the calibrated members and calibrated ensembles with optimal temperature values.

2020) on the validation data. We use the unbiased quadratic version of squared kernel calibration error (SKCE) with Laplacian kernel and kenel bandwidth chosen by median heuristic as one of the calibration error metrics(Widmann et al., 2019) . All three methods effectively improve the ensemble prediction calibration, with the dynamic approach yielding the best performance. We further investigate the impact of region numbers on the dynamic approach, as shown in Figure 3. It can be found that increasing the region number tends to improve the calibration performance, while requiring more parameters.

Finally, for the topology ensemble, weights were optimised using either maximum likelihood (Max LL) or area under curve (AUC) Zhong & Kwok (2013) (results in A.4). In Figure 2, the ensemble of calibrated structures is shown to be uncalibrated, with reliability curves typically slightly above the diagonal line. When the ensemble prediction is calibrated it can be seen that the calibration for the ensemble prediction is lower than the individual calibration errors in Table 1 ("post" lines).

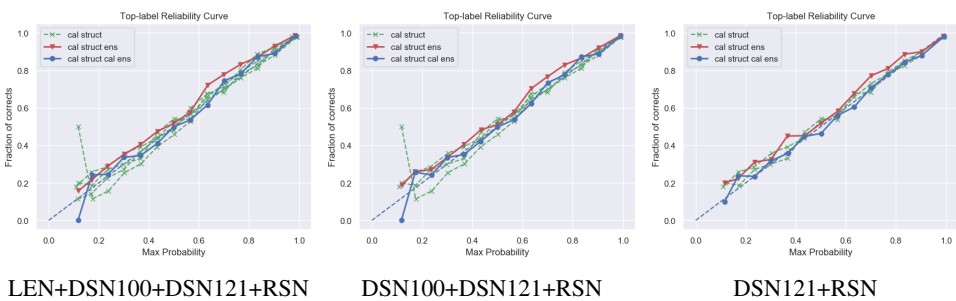

LEN+DSN100+DSN121+RSN        DSN100+DSN121+RSN        DSN121+RSN

Figure 2: Reliability curves of weighted combination of 4 calibrated structures, LEN, DSN100, DSN121 and RSN on CIFAR-100. The weightes are estimated by Max LL. Each structure is an ensemble of 10 models.

Table 1: Temperature calibration techniques on CIFAR-100, calibration parameters optimized to minimize ECE on validation set. In the "pre" mode, each member is scaled with one separate temperature. "dyn." denotes dynamic temperature scaling in post-combination mode using 6 region-based temperatures. Ranges indicate $\pm 2\sigma$.

| Model | Cal. | Acc.(%) | NLL | ACCE($10^{-4}$) | ACE($10^{-4}$) | ECCE($10^{-2}$) | ECE($10^{-2}$) | SKCE ($10^{-4}$) |
|---|---|---|---|---|---|---|---|---|
| LEN | — | 49.20 | 1.9741±0.0059 | 30.82±0.44 | 23.66±0.55 | 16.23±0.39 | 11.55±0.39 | **23.97±0.06** |
| | pre | 49.17 | 1.9641±0.0137 | 23.15±0.86 | 8.54±1.85 | 13.23±0.19 | 3.24±0.37 | 27.24±0.42 |
| | post | 49.20 | **1.9285±0.0068** | 21.72±0.61 | 5.73±1.24 | 13.22±0.23 | **2.19±0.45** | 28.41±0.44 |
| | dyn. | 49.20 | **1.9280±0.0107** | **21.19±0.73** | **4.45±1.66** | 12.86±0.28 | 2.33±1.05 | 28.81±0.30 |
| DSN 100 | — | 81.32 | 0.6699±0.0015 | 16.31±0.42 | 5.79±0.38 | 8.92±0.39 | 2.54±0.19 | **53.71±0.14** |
| | pre | 81.29 | 0.6912±0.0084 | 16.89±0.36 | 6.86±0.59 | 8.79±0.30 | 2.08±0.38 | 55.32±0.47 |
| | post | 81.32 | 0.6852±0.0080 | 16.73±0.38 | 6.29±0.68 | 8.56±0.25 | 1.83±0.32 | 57.64±1.25 |
| | dyn. | 81.32 | **0.6781±0.0058** | **16.11±0.60** | **4.94±1.03** | **8.41±0.35** | **1.31±0.53** | 57.17±0.69 |
| DSN 121 | — | 82.69 | **0.6314±0.0022** | 15.74±0.24 | 3.64±0.35 | **8.58±0.18** | **1.58±0.24** | 59.30±0.07 |
| | pre | 82.70 | **0.6312±0.0056** | 15.79±0.38 | 3.58±0.80 | **8.58±0.18** | 1.62±0.20 | 59.21±0.82 |
| | post | 82.69 | 0.6324±0.0044 | 15.81±0.43 | 3.76±0.65 | **8.57±0.17** | 1.56±0.23 | 59.61±1.30 |
| | dyn. | 82.69 | **0.6315±0.0041** | **15.63±0.43** | **3.26±0.34** | 8.65±0.29 | 1.71±0.18 | **57.87±0.53** |
| RSN | — | 83.45 | 0.6231±0.0023 | 16.95±0.20 | 7.31±0.26 | 9.28±0.26 | 3.22±0.19 | **57.01±0.14** |
| | pre | 83.41 | 0.6129±0.0018 | **15.41±0.43** | **2.52±0.63** | 8.75±0.17 | 1.88±0.26 | 60.67±0.70 |
| | post | 83.45 | 0.6118±0.0016 | **15.48±0.28** | 3.28±0.52 | 8.75±0.15 | 1.82±0.20 | 60.75±0.56 |
| | dyn. | 83.45 | **0.6097±0.0023** | 15.63±0.31 | 2.83±0.56 | **8.68±0.31** | **1.20±0.41** | 59.36±0.74 |

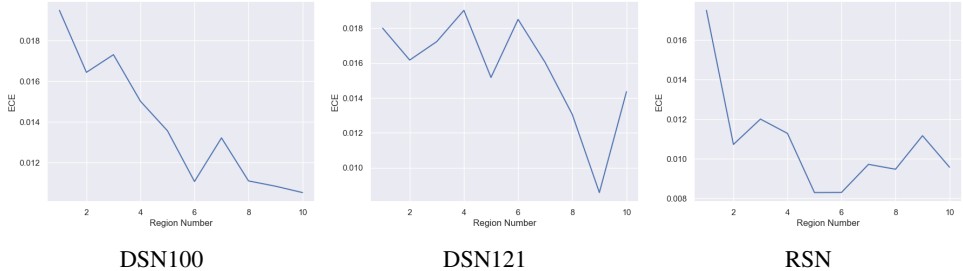

DSN100      DSN121      RSN

Figure 3: Impact of different region numbers on dynamic temperature annealing in calibration of ensembles of DSN100, DSN121 and RSN.

Table 2: Topology ensembles for CIFAR-100, optimal weights using ML estimation. Calibrations of each topology and ensemble using post-combination mode ("post" in Table 1).

| Weight Est. | Comb. Weight | | | | Acc. | Ens Cal. | NLL | ACE | ECE |
|---|---|---|---|---|---|---|---|---|---|
| | LEN | DSN100 | DSN121 | RSN | (%) | | | ($10^{-4}$) | ($10^{-2}$) |
| | 0.02 | 0.19 | 0.30 | 0.49 | 83.75 | — | 0.5766 | 4.97 | 2.24 |
| | | | | | | ✓ | 0.5698 | 1.42 | 1.20 |
| Max LL | — | 0.22 | 0.30 | 0.48 | 83.80 | — | 0.5741 | 3.74 | 2.00 |
| | | | | | | ✓ | 0.5714 | 1.52 | 1.29 |
| | — | — | 0.44 | 0.56 | 83.86 | — | 0.5816 | 3.64 | 2.06 |
| | | | | | | ✓ | 0.5801 | 2.36 | 1.35 |

## 5 CONCLUSIONS

State-of-the-art deep learning models often exhibit poor calibration performance. In this paper two aspects of calibration for these models are investigated: the theoretical definition of calibration and associated attributes for both general and top-label calibration; and the application of calibration to ensemble methods that are often used in deep-learning approaches for improved performance and uncertainty estimation. It is shown that calibrating members of the ensemble is not sufficient to ensure that the ensemble prediction is itself calibrated. The resulting ensemble predictions will be under-confident, requiring calibration functions to be optimised for the ensemble prediction, rather than ensemble members. These theoretical results are backed-up by empirical analysis on CIFAR-100 deep-learning models, with ensemble performance being robust to poorly calibrated ensemble members but requiring calibration even with well calibrated members.

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

## A APPENDIX

### A.1 THEORETICAL PROOF

**Proposition 3.** If all members are calibrated and the regions are the same, i.e., for different members $\boldsymbol{\theta}^{(m)}$ and $\boldsymbol{\theta}^{(m')}$

$$\mathcal{R}_j^p(\boldsymbol{\theta}^{(m)}, \epsilon) = \mathcal{R}_j^p(\boldsymbol{\theta}^{(m')}, \epsilon) \quad \forall p, \omega_j, \quad \epsilon \to 0$$

then the ensemble is also calibrated on the same regions

$$\int_{\boldsymbol{x} \in \mathcal{R}_j^p(\boldsymbol{\Theta}, \epsilon)} \mathrm{P}(\omega_j|\boldsymbol{x}; \boldsymbol{\Theta})\mathrm{p}(\boldsymbol{x})\mathrm{d}\boldsymbol{x} = \int_{\boldsymbol{x} \in \mathcal{R}_j^p(\boldsymbol{\Theta}, \epsilon)} \mathrm{P}(\omega_j|\boldsymbol{x})\mathrm{p}(\boldsymbol{x})\mathrm{d}\boldsymbol{x}, \quad \forall p, \omega_j, \quad \epsilon \to 0$$

*Proof.* If

$$\mathcal{R}_j^p(\boldsymbol{\theta}^{(m)}, \epsilon) = \mathcal{R}_j^p(\boldsymbol{\theta}^{(m')}, \epsilon) \quad \forall p, \omega_j, \quad \epsilon \to 0$$

The ensemble is also calibrated and the regions are the same:

$$\mathcal{R}_j^p(\boldsymbol{\Theta}, \epsilon) = \left\{ \boldsymbol{x} \,\middle|\, \left| \frac{1}{M} \sum_{m=1}^M \mathrm{P}(\omega_j|\boldsymbol{x}; \boldsymbol{\theta}^{(m)}) - p \right| \le \epsilon \right\} = \mathcal{R}_j^p(\boldsymbol{\theta}^{(m)}) \quad \forall p, \omega_j, \quad \epsilon \to 0 \tag{30}$$

$$\int_{\boldsymbol{x} \in \mathcal{R}_j^p(\boldsymbol{\Theta}, \epsilon)} \frac{1}{M} \sum_{m=1}^M \mathrm{P}(\omega_j|\boldsymbol{x}; \boldsymbol{\theta}^{(m)})\mathrm{p}(\boldsymbol{x})\mathrm{d}\boldsymbol{x} = \frac{1}{M} \sum_{m=1}^M \int_{\boldsymbol{x} \in \mathcal{R}_j^p(\boldsymbol{\theta}^{(m)}, \epsilon)} \mathrm{P}(\omega_j|\boldsymbol{x}; \boldsymbol{\theta}^{(m)})\mathrm{p}(\boldsymbol{x})\mathrm{d}\boldsymbol{x}$$

$$= \frac{1}{M} \sum_{m=1}^M \int_{\boldsymbol{x} \in \mathcal{R}_j^p(\boldsymbol{\theta}^{(m)}, \epsilon)} \mathrm{P}(\omega_j|\boldsymbol{x})\mathrm{p}(\boldsymbol{x})\mathrm{d}\boldsymbol{x}$$

$$= \int_{\boldsymbol{x} \in \mathcal{R}_j^p(\boldsymbol{\theta}^{(m)}, \epsilon)} \mathrm{P}(\omega_j|\boldsymbol{x})\mathrm{p}(\boldsymbol{x})\mathrm{d}\boldsymbol{x}$$

$$= \int_{\boldsymbol{x} \in \mathcal{R}_j^p(\boldsymbol{\Theta}, \epsilon)} \mathrm{P}(\omega_j|\boldsymbol{x})\mathrm{p}(\boldsymbol{x})\mathrm{d}\boldsymbol{x}$$

□

**Proposition 4.** When class number $K > 2$, if all members are globally top-label calibrated, then the ensemble is not necessarily global top-label calibrated.

*Proof.* Assume globally top-label calibrated members imply globally top-label calibrated ensemble, that is, given

$$\frac{1}{N} \sum_{i=1}^N \mathrm{P}(\hat{y}_m^{(i)}|\boldsymbol{x}^{(i)}; \boldsymbol{\theta}^{(m)}) = \frac{1}{N} \sum_{i=1}^N \delta(y^{(i)}, \hat{y}_m^{(i)}), \quad m = 1, ..., M \tag{31}$$

the following is true

$$\frac{1}{N} \sum_{i=1}^N \mathrm{P}(\hat{y}_{\mathrm{E}}^{(i)}|\boldsymbol{x}^{(i)}; \boldsymbol{\Theta}) = \frac{1}{N} \sum_{i=1}^N \delta(y^{(i)}, \hat{y}_{\mathrm{E}}^{(i)}) \tag{32}$$

If $\exists n, \tilde{m}, \tau > 0$, such that $\hat{y}_{\mathrm{E}}^{(n)} \ne \hat{y}_{\tilde{m}}^{(n)}$, then it is possible to write

$$\mathrm{P}(\hat{y}_{\mathrm{E}}^{(n)}|\boldsymbol{x}^{(n)}; \boldsymbol{\Theta}) = \left( \frac{1}{M} \sum_{m \ne \tilde{m}} \mathrm{P}(\hat{y}_{\mathrm{E}}^{(n)}|\boldsymbol{x}^{(n)}; \boldsymbol{\theta}^{(m)}) \right) + \frac{1}{M} \mathrm{P}(\hat{y}_{\mathrm{E}}^{(n)}|\boldsymbol{x}^{(n)}; \boldsymbol{\theta}^{(\tilde{m})}) \tag{33}$$

For top-label calibration there are no constraints on the second term in Eq. (33) as it is not the top-label for model $\boldsymbol{\theta}^{(\tilde{m})}$. Thus there are a set of models that satisfy the top-label calibration constraints for member $\tilde{m}$ that only need to satisfy the following constraints

$$0 \le \mathrm{P}(\hat{y}_{\mathrm{E}}^{(n)}|\boldsymbol{x}^{(n)}; \boldsymbol{\theta}^{(\tilde{m})}) < \mathrm{P}(\hat{y}_{\tilde{m}}^{(n)}|\boldsymbol{x}^{(n)}; \boldsymbol{\theta}^{(\tilde{m})}) \le 1 \tag{34}$$

and the standard sum-to-one constraint over all classes. Consider replacing member $\tilde{m}$ of the ensemble with a member having parameters $\tilde{\boldsymbol{\theta}}^{(\tilde{m})}$, to yield $\tilde{\boldsymbol{\Theta}}$, that satisfies

$$\max_\omega \left\{ \text{P}(\omega|\boldsymbol{x}^{(n)}; \tilde{\boldsymbol{\theta}}^{(\tilde{m})}) \right\} = \max_\omega \left\{ \text{P}(\omega|\boldsymbol{x}^{(n)}; \boldsymbol{\theta}^{(\tilde{m})}) \right\} = \hat{y}_{\tilde{m}}^{(n)} \tag{35}$$

$$\text{P}(\hat{y}_{\tilde{m}}^{(n)}|\boldsymbol{x}^{(n)}; \tilde{\boldsymbol{\theta}}^{(\tilde{m})}) = \text{P}(\hat{y}_{\tilde{m}}^{(n)}|\boldsymbol{x}^{(n)}; \boldsymbol{\theta}^{(\tilde{m})}) \tag{36}$$

$$\text{P}(\hat{y}_{\text{E}}^{(n)}|\boldsymbol{x}^{(n)}; \tilde{\boldsymbol{\theta}}^{(\tilde{m})}) = \text{P}(\hat{y}_{\text{E}}^{(n)}|\boldsymbol{x}^{(n)}; \boldsymbol{\theta}^{(\tilde{m})}) + \tau \tag{37}$$

where $\tau > 0$, and the standard sum-to-one constraint is satisfied, and all other predictions are unaltered. This results in the following constraints

$$\max_\omega \left\{ \text{P}(\omega|\boldsymbol{x}^{(n)}; \tilde{\boldsymbol{\Theta}}) \right\} = \max_\omega \left\{ \text{P}(\omega|\boldsymbol{x}^{(n)}; \boldsymbol{\Theta}) \right\} = \hat{y}_{\text{E}}^{(n)} \tag{38}$$

$$\text{P}(\hat{y}_{\text{E}}^{(n)}|\boldsymbol{x}^{(n)}; \boldsymbol{\Theta}) < \text{P}(\hat{y}_{\text{E}}^{(n)}|\boldsymbol{x}^{(n)}; \tilde{\boldsymbol{\Theta}}) \tag{39}$$

The accuracy of the two ensembles $\boldsymbol{\Theta}$ and $\tilde{\boldsymbol{\Theta}}$ are the same from Eq. (38), but the probabilities associated with those predictions cannot be the same from Eq. (39), so both ensemble predictions cannot be calibrated, as assuming that the ensemble prediction for $\boldsymbol{\Theta}$ is calibrated

$$\frac{1}{N} \sum_{i=1}^{N} \text{P}(\hat{y}_{\text{E}}^{(i)}|\boldsymbol{x}^{(i)}; \tilde{\boldsymbol{\Theta}}) > \frac{1}{N} \sum_{i=1}^{N} \text{P}(\hat{y}_{\text{E}}^{(i)}|\boldsymbol{x}^{(i)}; \boldsymbol{\Theta}) = \frac{1}{N} \sum_{i=1}^{N} \delta(y^{(i)}, \hat{y}_{\text{E}}^{(i)}) \tag{40}$$

Hence there are multiple values of $\text{P}(\hat{y}_{\text{E}}^{(n)}|\boldsymbol{x}^{(n)}; \boldsymbol{\Theta})$ for which all the models satisfy the top-calibration constraints, but these cannot all be consistent with Eq. (40). For the situation where there is no sample or model where $\hat{y}_{\text{E}}^{(n)} \neq \hat{y}_{\tilde{m}}^{(n)}$ then the predictions for all models for all samples are the same as the ensemble prediction, so by definition there can be no performance gain.

$\square$

## A.2 Global General Calibration and Top-label Calibration

To demonstrate the differences between global top-label calibration and global calibration, a set of ensemble member predictions were generated using Algorithm 1, this ensures that the predictions are perfectly calibrated. Since the member predictions are perfectly calibrated, the ensemble members will be globally calibrated. Figure 4 (a) shows the performance in terms of ACE of the ensemble prediction as the value of $\epsilon$ increases, note when $\epsilon = 1$ this is a global calibration version of ACE. It can be seen that as $\epsilon$ increases ACE decreases, and for the global case reduces to zero for the ensemble predictions as the theory states.

In terms of top-label calibration, as the ensemble members are perfectly calibrated, they will again be global top-label calibrated. This is illustrated in Figure 4 (b) where ECE is zero for all ensemble members. For top-label calibration the value of ECE does not decrease to zero as the $\epsilon \to 1$, again as the theory states. This is because the underlying probability regions associated with each of the members of the ensemble are different. Hence, even for perfectly calibrated ensemble members, the ensemble prediction is not global top-label calibrated.

## A.3 Toy Datasets

**Example 1.** In this example, we show the difference between all-label calibration and top-label calibration which consider the different regions in Eq. (3) and Eq. (4).

Assuming $\text{p}(\boldsymbol{x}) \propto 1$, the whole input space $\mathcal{X}$ is consisted of three regions $\mathcal{R}_1$, $\mathcal{R}_2$ and $\mathcal{R}_3$, and

$$\int_{\boldsymbol{x} \in R_1} \text{p}(\boldsymbol{x})d\boldsymbol{x} = \int_{\boldsymbol{x} \in R_2} \text{p}(\boldsymbol{x})d\boldsymbol{x} = \int_{\boldsymbol{x} \in R_3} \text{p}(\boldsymbol{x})d\boldsymbol{x}. \tag{41}$$

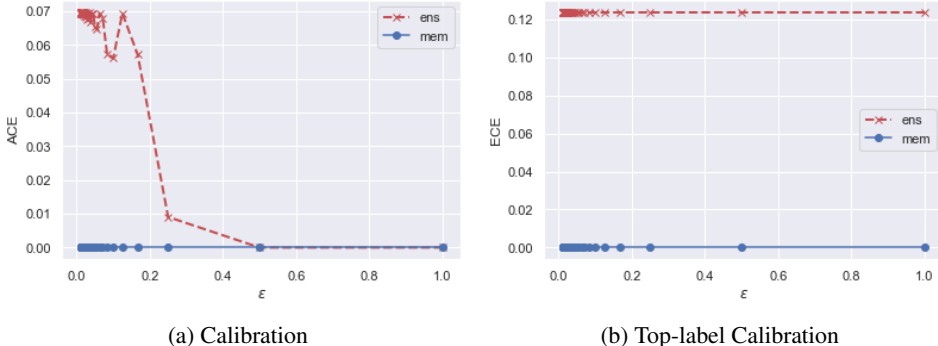

(a) Calibration            (b) Top-label Calibration

Figure 4: ACE (calibration) and ECE (top-label calibration) of a perfectly calibrated set of ensemble members, as the value of $\epsilon$ varies.

The corresponding system prediction $\hat{P}$ and the true distribution $P$ is:

$$
\hat{P} = \left( P(\omega_j | \boldsymbol{x} \in \mathcal{R}_r; \boldsymbol{\theta}) \right) = \begin{matrix} & \omega_1 & \omega_2 & \omega_3 & \omega_4 & \\ \begin{pmatrix} & 0.5 & \underline{0.4} & 0.05 & 0.05 \\ & 0.3 & \underline{0.4} & 0.2 & 0.1 \\ & 0.3 & 0.3 & 0.35 & 0.05 \end{pmatrix} & \begin{matrix} \mathcal{R}_1 \\ \mathcal{R}_2 \\ \mathcal{R}_3 \end{matrix} \end{matrix} \tag{42}
$$

$$
P = \begin{matrix} & \omega_1 & \omega_2 & \omega_3 & \omega_4 & \\ \begin{pmatrix} & 0.5 & \underline{0.4 - \tau} & 0.05 & 0.05 + \tau \\ & 0.3 - \tau & \underline{0.4 + \tau} & 0.2 & 0.1 \\ & 0.3 + \tau & 0.3 & 0.35 & 0.05 - \tau \end{pmatrix} & \begin{matrix} \mathcal{R}_1 \\ \mathcal{R}_2 \\ \mathcal{R}_3 \end{matrix} \end{matrix} , \quad \tau > 0 \tag{43}
$$

It can be verified that

$$
\int_{\boldsymbol{x} \in \mathcal{R}_j^p(\boldsymbol{\theta}, \epsilon)} P(\omega_j | \boldsymbol{x}, \boldsymbol{\theta}) p(\boldsymbol{x}) \mathrm{d}\boldsymbol{x} = \int_{\boldsymbol{x} \in \mathcal{R}_j^p(\boldsymbol{\theta}, \epsilon)} P(\omega_j | \boldsymbol{x}) p(\boldsymbol{x}) \mathrm{d}\boldsymbol{x}, \quad \forall \omega_j, p, \epsilon \to 0 \tag{44}
$$

However, when $p = 0.4, j = 2$

$$
\int_{\boldsymbol{x} \in \tilde{\mathcal{R}}_j^p(\boldsymbol{\theta}, \epsilon)} P(\omega_j | \boldsymbol{x}, \boldsymbol{\theta}) p(\boldsymbol{x}) \mathrm{d}\boldsymbol{x} \neq \int_{\boldsymbol{x} \in \tilde{\mathcal{R}}_j^p(\boldsymbol{\theta}, \epsilon)} P(\omega_j | \boldsymbol{x}, \boldsymbol{\theta}) p(\boldsymbol{x}) \mathrm{d}\boldsymbol{x}, \quad \epsilon \to 0 \tag{45}
$$

**Example 2.** This example shows that combination of calibrated members yields uncalibrated ensemble. Algorithm 1 generates the true data distribution $p$ by sampling from a Dirichlet distribution with equal concentration parameters of 1. To generate the member predictions, the $N$ samples are randomly assigned to $N/b$ bins with size of $b$. In each bin, the member predictions $\hat{p}$ are equally the average of true data distribution of the associated samples. This ensures that for each region, Eq. (2) holds for each member. However, regions for different member are different due to the random assignment. Therefore, the corresponding ensemble is not automatically calibrated. Figure 4 shows that the ensemble is uncalibrated with ACE of 0.0697.

**Example 3.** In this example, we show that for a finite number of samples, the regions $\mathcal{S}_j^p(\boldsymbol{\theta}, \epsilon)$ and $\tilde{\mathcal{S}}_j^p(\boldsymbol{\theta}, \epsilon)$ derived from the samples is different from the theoretical regions, leading to difference between theoretical calibration error measures and the values estimated from the finite samples. Algorithm 2 generates data with difference between finite sample-based calibration error and theoretical error. The theoretical ACE of the predicted probabilities in Algorithm 2 is $\frac{9}{32}$. However, the finite sample-based ACE is 0.

The true data distribution is $P(\omega_j | \boldsymbol{x}) = \frac{1}{4}, p(\boldsymbol{x}) \propto 1$. The samples in $\mathcal{D}$ are assigned to bins of size 2. The type of bin that a sample is assigned to determines the predicted probability of sample. Considering each class $\omega_j$, there are three types of bins:
$\mathbf{B}_{p=1}$: both samples belong to class $\omega_j$.
$\mathbf{B}_{p=0.5}$: only one sample is class $\omega_j$.

---

**Algorithm 1:** Algorithm for generating calibrated members that yield uncalibrated ensemble

---

**Result:** $\{p^{(i)}\}_{i=1}^{N}, \{\hat{p}^{(i)}, \hat{y}^{(i)}\}_{i=1}^{N}$

$b = 2$; //bin size, number of samples in one bin;

$K = 4$; //number of classes;

$M = 10$; //number of members;

$N = 1000000$; //number of samples;

$p^{(i)} \sim \text{Dirichlet}(\boldsymbol{\alpha} = \mathbf{1}), i = 1, ..., N$; //true data distribution, sampled from Dirichlet
distribution with equal concentration parameters of 1;

$\boldsymbol{I} = [1, 2, ..., N]$; // index vector;

**for** $m$ *in* $[1, ..., M]$ **do**

    $\tilde{\boldsymbol{I}} \leftarrow \text{shuffle}(\boldsymbol{I})$;

    **for** $j$ *in* $[1, ..., \lceil N/b \rceil]$ **do**

        $\boldsymbol{B}_j = (b * (j - 1), \min\{b * j, N\}]$;

        $\hat{\boldsymbol{p}}_{\boldsymbol{B}_j} = \frac{1}{b} \sum_{l \in B_j} \boldsymbol{p}^{(\tilde{I}_l)}$;

    **end**

    **for** $i$ *in* $[1, ..., N]$ **do**

        $j = \lceil \frac{i}{b} \rceil$;

        $\hat{\boldsymbol{p}}^{(\tilde{I}_i, m)} = \hat{\boldsymbol{p}}_{\boldsymbol{B}_j}$;

    **end**

**end**

---

$\mathbf{B}_{p=0}$: both samples are not class $\omega_j$.

$$\text{P}(\boldsymbol{x} \in \mathbf{B}_1) = \frac{1}{16}, \ \ \text{P}(\boldsymbol{x} \in \mathbf{B}_{0.5}) = \frac{6}{16}, \ \ \text{P}(\boldsymbol{x} \in \mathbf{B}_0) = \frac{9}{16} \tag{46}$$

then

$$\int_{\boldsymbol{x} \in \mathcal{R}_j^p(\boldsymbol{\theta}, 0)} \big( \text{P}(\omega_j | \boldsymbol{x}; \boldsymbol{\theta}) - \text{P}(\omega_j | \boldsymbol{x}) \big) \text{p}(\boldsymbol{x}) d\boldsymbol{x} \ = \ \int_{\boldsymbol{x} \in \mathcal{R}_j^p(\boldsymbol{\theta}, 0)} \big( p - \frac{1}{4} \big) \text{p}(\boldsymbol{x}) d\boldsymbol{x} \tag{47}$$

$$= \ (p - \frac{1}{4}) \text{P}(\boldsymbol{x} \in \mathbf{B}_p) \tag{48}$$

therefore,

$$\text{ACE}(\boldsymbol{\theta}) \ = \ \frac{1}{4} \int_0^1 \Big| \sum_{j=1}^{4} \int_{\boldsymbol{x} \in \mathcal{R}_j^p(\boldsymbol{\theta}, 0)} \big( \text{P}(\omega_j | \boldsymbol{x}; \boldsymbol{\theta}) - \text{P}(\omega_j | \boldsymbol{x}) \big) \text{p}(\boldsymbol{x}) d\boldsymbol{x} \Big| dp \tag{49}$$

$$= \ \sum_{p \in \{0, 0.5, 1\}} \Big| (p - \frac{1}{4}) \text{P}(\boldsymbol{x} \in \mathbf{B}_p) \Big| = \frac{9}{32} \tag{50}$$

## A.4 ADDITIONAL EXPERIMENTAL RESULTS

In this section, we show some comparison experiments to the empirical results in Section 4.3. We conducted experiments on CIFAR-100 and CIFAR-10 dataset. Table 3 and 4 display the performance of inidividual models, LeNet, DenseNet 100, DenseNet 121, and wide ResNet 28-10. All systems are trained with data augmentation of random cropping and horizontal flipping, and simple mean/std normalization. The original training/test images in CIFAR datasets is 50,000/10,000. We hold out 5,000 images from training set as validation set (10%) for temperature and combination weights optimization, this leads to a slight accuracy degradation compared to training with all 50,000 images. We have 10 runs of our experiment to obtain the deviations.

In Section 4.3, we presented ensemble calibration on CIFAR-100. The counterpart on CIFAR-10 is given in Table 5. Other than separately specified, all sample based evaluation criteria in this paper use 15 bins following previous literature (Guo et al., 2017). The temperatures in pre-, post- and dynamic post-combination modes are optimized on the validation set by minimizing ECE (Liang et al.,

---

**Algorithm 2:** Algorithm for generating data with difference between finite sample-based ACE and theoretical ACE.

---

**Result:** $\{\hat{\boldsymbol{p}}^{(i)}, \hat{y}^{(i)}\}_{i=1}^{N}$
$b = 2$; //bin size, number of samples in one bin;
$K = 4$; //number of classes;
$\boldsymbol{g} = [g_1, ..., g_N]$; //vector of ground-truth labels , where $g \in \{1, ..., K\}$. Numbers of labels for different classes are equal;
$\boldsymbol{I} = [1, 2, ..., N]$; // index vector;
$\tilde{\boldsymbol{I}} \leftarrow \text{shuffle}(\boldsymbol{I})$;
**for** $j$ *in* $[1, ..., \lceil N/b \rceil]$ **do**
  $\quad \boldsymbol{B}_j = (b * (j - 1), \min\{b * j, N\}]$;
  $\quad \hat{\boldsymbol{p}}_{\boldsymbol{B}_j} = \frac{1}{b} \sum_{l \in \boldsymbol{B}_j} [\delta(1, g_{\tilde{I}_l}), \delta(2, g_{\tilde{I}_l}), ..., \delta(K, g_{\tilde{I}_l})]$;
**end**
**for** $i$ *in* $[1, ..., N]$ **do**
  $\quad j = \lceil \frac{i}{b} \rceil$;
  $\quad \hat{\boldsymbol{p}}^{(\tilde{I}_i)} = \hat{\boldsymbol{p}}_{\boldsymbol{B}_j}$;
**end**

---

2020), using SGD with learning rate of 0.1 for 400 iterations. It can be observed that combination of DenseNet and ResNet improves the calibration performance, while combination of LeNet doesn't help. This is because LeNet is not as over-confident as DenseNet and ResNet (as shown in Figure 1). Hence the simple ensemble combination doesn't help, but aggravates the calibration. The three temperature-based calibration methods effectively improve the system calibration performance on CIFAR-10 as well.

Table 6 gives the ensemble combination based on AUC weights (Zhong & Kwok, 2013). The AUC weights are much even than the Max LL weights in Table 2. The structures combined are first calibrated, nevertheless, applying post-combination calibration to the ensemble obtains gains. We evaluated post-combination for topology combination in Table 7. The post- and dynamic post-combination methods are applied to calibrate the topologies and the topology ensemble. The dynamic temperature method shows clear advantage in obtaining calibrated ensemble of mutiple topologies.

Table 3: Individual model performance on CIFAR-100. Full data training (100%Train) and training with 5,000 images held out (90%Train) are presented. * denotes results from the original papers.

| Model | %Train | Acc.(%) (%) | NLL | ACCE $(10^{-4})$ | ACE $(10^{-4})$ | ECCE $(10^{-2})$ | ECE $(10^{-2})$ |
|---|---|---|---|---|---|---|---|
| LEN | 100 | 42.47 | 2.2730 | 21.07 | 7.62 | 13.05 | 1.82 |
| | 90 | 42.12 | 2.2652 | 22.80 | 10.11 | 13.20 | 2.86 |
| DSN 100 | 100* | 76.21 | - | - | - | - | - |
| | 100 | 76.71 | 1.1022 | 30.54 | 25.85 | 14.53 | 12.65 |
| | 90 | 76.00 | 1.0591 | 30.22 | 24.55 | 13.91 | 11.72 |
| DSN 121 | 100 | 80.00 | 0.8438 | 23.39 | 17.29 | 11.52 | 8.59 |
| | 90 | 78.87 | 0.8981 | 24.83 | 19.00 | 11.99 | 9.41 |
| RSN | 100* | 80.75 | - | - | - | - | - |
| | 100 | 80.99 | 0.7711 | 18.67 | 8.68 | 10.25 | 5.02 |
| | 90 | 80.06 | 0.7985 | 18.72 | 8.83 | 10.23 | 4.76 |

Table 4: Individual model performance on CIFAR-10. Full data training (100%Train) and training with 5,000 images held out (90%Train) are presented. * denotes results from the original papers.

| Model | %Train | Acc.(%) (%) | NLL | ACCE ($10^{-4}$) | ACE ($10^{-4}$) | ECCE ($10^{-2}$) | ECE ($10^{-2}$) |
|---|---|---|---|---|---|---|---|
| LEN | 100 | 75.07 | 0.7329 | 60.85 | 23.97 | 3.55 | 1.24 |
| | 90 | 75.31 | 0.7280 | 77.17 | 44.49 | 4.16 | 2.09 |
| DSN 100 | 100* | 94.23 | - | - | - | - | - |
| | 100 | 95.39 | 0.2003 | 61.04 | 54.74 | 3.04 | 2.77 |
| | 90 | 94.92 | 0.2181 | 65.50 | 60.54 | 3.27 | 2.99 |
| DSN 121 | 100 | 95.66 | 0.1832 | 61.56 | 56.63 | 3.01 | 2.87 |
| | 90 | 95.53 | 0.1993 | 62.79 | 57.16 | 3.06 | 2.89 |
| RSN 28 | 100* | 96.00 | - | - | - | - | - |
| | 100 | 95.98 | 0.1524 | 51.67 | 46.70 | 2.61 | 2.35 |
| | 90 | 95.52 | 0.1692 | 58.07 | 52.70 | 2.89 | 2.62 |

Table 5: Calibration using temperature annealing on CIFAR-10. The temperatures are optimized to minimize ECE on validation set. In the 'pre' mode, each member is scaled with one separate temperature. 'dyn.' denotes dynamic temperature scaling in post-combination mode using 6 region-based temperatures. The three structures investigated are LeNet 5, DenseNet 121, DenseNet 100 and Wide ResNet 28. Ranges indicate $\pm 2\sigma$.

| Model | Cal. | Acc.(%) | NLL | ACCE($10^{-4}$) | ACE($10^{-4}$) | ECCE($10^{-2}$) | ECE($10^{-2}$) | SKCE($10^{-4}$) |
|---|---|---|---|---|---|---|---|---|
| LEN | — | 80.11 | 0.6067±0.0031 | 187.73±3.22 | 181.18±1.86 | 9.55±0.26 | 9.18±0.17 | **447.50±1.78** |
| | pre | 80.04 | 0.5915±0.0062 | 76.68±6.11 | 53.50±5.10 | 4.01±0.24 | 1.91±0.34 | 546.65±5.03 |
| | post | 80.11 | **0.5664±0.0033** | 57.84±3.90 | **24.00±4.62** | **3.66±0.21** | **1.15±0.31** | 568.86±6.55 |
| | dyn. | 80.11 | 0.5718±0.0071 | **57.79±5.58** | 25.45±8.26 | 3.74±0.36 | 1.29±0.52 | 568.94±7.07 |
| DSN 100 | — | 96.19 | 0.1230±0.0013 | 28.01±2.23 | 13.30±1.95 | 1.47±0.10 | 0.55±0.13 | 871.27±0.72 |
| | pre | 96.19 | 0.1227±0.0013 | 27.92±2.56 | 13.00±2.17 | 1.46±0.10 | 0.54±0.12 | 870.62±0.91 |
| | post | 96.19 | 0.1228±0.0015 | 28.27±2.94 | 13.27±1.80 | 1.47±0.13 | 0.55±0.13 | 870.24±3.31 |
| | dyn. | 96.19 | **0.1224±0.0015** | **27.15±2.58** | **11.16±1.93** | **1.43±0.13** | **0.45±0.11** | 867.59±3.52 |
| DSN 121 | — | 96.35 | 0.1202±0.0010 | 27.98±1.17 | 15.31±1.46 | 1.43±0.12 | 0.69±0.14 | 895.02±0.59 |
| | pre | 96.35 | 0.1198±0.0012 | 27.85±1.33 | 15.08±1.46 | 1.42±0.12 | 0.67±0.13 | 894.55±0.98 |
| | post | 96.35 | 0.1194±0.0027 | **27.37±2.67** | **14.54±2.73** | 1.41±0.14 | 0.64±0.21 | 892.48±8.64 |
| | dyn. | 96.35 | **0.1191±0.0012** | 27.66±1.56 | 14.60±2.42 | 1.42±0.15 | 0.66±0.14 | 894.06±3.28 |
| RSN | — | 96.51 | 0.1081±0.0008 | 25.14±1.34 | 11.73±1.39 | 1.37±0.10 | 0.58±0.10 | 894.64±1.07 |
| | pre | 96.51 | 0.1073±0.0009 | 24.54±2.04 | 10.01±1.12 | 1.32±0.10 | 0.50±0.11 | 891.23±1.80 |
| | post | 96.51 | **0.1069±0.0007** | 23.96±1.35 | 8.83±0.96 | 1.32±0.15 | 0.50±0.09 | **881.57±4.60** |
| | dyn. | 96.51 | **0.1070±0.0007** | **23.84±1.84** | **8.57±1.54** | **1.30±0.15** | **0.47±0.13** | 883.58±5.70 |

Table 6: Topology ensembles for CIFAR-100, optimal weights based on AUC. Calibrations of each topology and ensemble using post-combination mode ("post" in Table 1).

| Weight Est. | Comb. Weight | | | | Acc. (%) | Ens Cal. | NLL | ACE ($10^{-4}$) | ECE ($10^{-2}$) |
|---|---|---|---|---|---|---|---|---|---|
| | LEN | DSN100 | DSN121 | RSN | | | | | |
| AUC | 0.20 | 0.26 | 0.27 | 0.27 | 83.30 | — | 0.6649 | 20.84 | 10.07 |
| | | | | | | ✓ | 0.6096 | 2.60 | 2.00 |
| | — | 0.33 | 0.33 | 0.34 | 83.55 | — | 0.5776 | 3.67 | 1.89 |
| | | | | | | ✓ | 0.5787 | 3.03 | 1.38 |
| | — | — | 0.49 | 0.51 | 83.78 | — | 0.5820 | 3.54 | 2.03 |
| | | | | | | ✓ | 0.5805 | 2.30 | 1.34 |

Table 7: Topology combination on CIFAR-100. Dynamic mode uses 6 region-based temperatures.

| | Comb. Weight | | | Struct cal. | Ens Cal. | Acc.(%) | NLL | ACE | ECE |
|---|---|---|---|---|---|---|---|---|---|
| LEN | DSN100 | DSN121 | RSN | | | | | | |
| - | 0.33 | 0.33 | 0.33 | - | - | 83.54 | 0.5846 | 0.0007 | 0.0334 |
| - | 0.33 | 0.33 | 0.33 | - | post | 83.54 | 0.5769 | 0.0002 | 0.0132 |
| - | 0.33 | 0.33 | 0.33 | - | dyn. | 83.54 | 0.5751 | 0.0002 | 0.0099 |
| - | 0.33 | 0.33 | 0.33 | post | - | 83.57 | 0.5779 | 0.0004 | 0.0189 |
| - | 0.33 | 0.33 | 0.33 | post | post | 83.57 | 0.5778 | 0.0002 | 0.0118 |
| - | 0.33 | 0.33 | 0.33 | dyn. | - | 83.59 | 0.5771 | 0.0004 | 0.0211 |
| - | 0.33 | 0.33 | 0.33 | dyn. | dyn. | 83.59 | 0.5730 | 0.0002 | 0.0112 |
| - | 0.22 | 0.39 | 0.39 | - | - | 83.78 | 0.5822 | 0.0007 | 0.0327 |
| - | 0.22 | 0.39 | 0.39 | - | post | 83.78 | 0.5724 | 0.0002 | 0.0135 |
| - | 0.22 | 0.39 | 0.39 | - | dyn. | 83.78 | 0.5705 | 0.0002 | 0.0113 |
| - | 0.22 | 0.30 | 0.48 | post | - | 83.80 | 0.5741 | 0.0004 | 0.0200 |
| - | 0.22 | 0.30 | 0.48 | post | post | 83.80 | 0.5714 | 0.0002 | 0.0129 |
| - | 0.23 | 0.31 | 0.46 | dyn. | - | 83.75 | 0.5741 | 0.0004 | 0.0211 |
| - | 0.23 | 0.31 | 0.46 | dyn. | dyn. | 83.75 | 0.5700 | 0.0002 | 0.0153 |
| 0.25 | 0.25 | 0.25 | 0.25 | - | - | 83.34 | 0.7238 | 0.0032 | 0.1551 |
| 0.25 | 0.25 | 0.25 | 0.25 | - | post | 83.34 | 0.6043 | 0.0003 | 0.0196 |
| 0.25 | 0.25 | 0.25 | 0.25 | - | dyn. | 83.34 | 0.6027 | 0.0002 | 0.0122 |
| 0.25 | 0.25 | 0.25 | 0.25 | post | - | 83.25 | 0.6947 | 0.0025 | 0.1224 |
| 0.25 | 0.25 | 0.25 | 0.25 | post | post | 83.25 | 0.6262 | 0.0004 | 0.0218 |
| 0.25 | 0.25 | 0.25 | 0.25 | dyn. | - | 83.22 | 0.6978 | 0.0026 | 0.1258 |
| 0.25 | 0.25 | 0.25 | 0.25 | dyn. | dyn. | 83.22 | 0.6184 | 0.0002 | 0.0137 |
| 0.01 | 0.19 | 0.42 | 0.38 | - | - | 83.74 | 0.5859 | 0.0008 | 0.0374 |
| 0.01 | 0.19 | 0.42 | 0.38 | - | post | 83.74 | 0.5710 | 0.0001 | 0.0114 |
| 0.01 | 0.19 | 0.42 | 0.38 | - | dyn. | 83.74 | 0.5703 | 0.0001 | 0.0085 |
| 0.02 | 0.19 | 0.30 | 0.49 | post | - | 83.75 | 0.5766 | 0.0005 | 0.0224 |
| 0.02 | 0.19 | 0.30 | 0.49 | post | post | 83.75 | 0.5698 | 0.0001 | 0.0120 |
| 0.02 | 0.21 | 0.30 | 0.47 | dyn. | - | 83.77 | 0.5784 | 0.0006 | 0.0282 |
| 0.02 | 0.21 | 0.30 | 0.47 | dyn. | dyn. | 83.77 | 0.5698 | 0.0002 | 0.0167 |

