# OpenReview forum: "Should Ensemble Members Be Calibrated?"
_ICLR.cc/2021/Conference — Reject_

### Official Review · AnonReviewer2 · 2020-10-26

**Rating:** 5
**Confidence:** 3

**Review:**

- In general, my opinion is aligned with AnonReviewer1 the theory and the empirical contribution do not feel sufficient.
- I also agree with AnonReviewer3  and AnonReviewer4 but feel less excited about the prons and more worried about the cons.

At this point, I'm not against the acceptance of the paper, although I'm still staying on the rejection side. I'm increasing my score because we are at least talking about a borderline.

--------------------------

Summary:

The paper study calibration of ensembles of DNNs and its relation to the calibration of individual members of ensembles. The work demonstrates that i) members of an ensemble should not be calibrated, but the final ensemble may require calibration (especially if members of an ensemble are calibrated) ii) provide theoretical results to support the statement iii) propose an adaptive calibration scheme (dynamic temperature scaling) that uses different temperatures based on the confidence of a model.


Concerns:

1) The main question of the paper "Should ensemble members be calibrated?" feels trivial, because the community is aware of the simple example that provides an answer. The Deep Ensembles [Lakshminarayanan2017] have miscalibrated members---conventional DNNs, but the predictions of an ensemble are, in-most-cases, calibrated. Thus the answer is "No".

2) The paper mostly is clearly written, but section 4.1 "Theoretical Analysis" is *extremely* hard to follow. Even though I re-read it many times, I'm still not sure if I understood it correctly. The most confusing part is the conclusion "In practice, there is no constraint that the ensemble prediction should be calibrated, thus ensemble prediction calibration is required even for top-label calibrated members.". It seems that no listed results were used to produce this statement.

3) The calibration of ensemble has been proposed in [Ashukha2020, 5 Discussion & Conclusion]. ("The resulting ensemble predictions ..., requiring calibration functions to be optimized for the ensemble prediction, rather than ensemble members.")

4) The two main contributions (4.1 Theoretical Analysis, 4.2 Temperature Annealing for Ensemble Calibration) feels not related, they are basically two independent topics packed in the one paper.

5) The empirical comparison exploits the calibrations score (e.g., ECE). ECE  is a biased estimate of true calibration with a different bias for each model, so it is not a valid metric to compare different models (see Vaicenavicius2019). The fact is even mentioned in the current paper ("It should be noted that for finite number of samples ... ") but still is ignored in the empirical study.

What I suggest is to use the squared kernel calibration error (SKCE) proposed in [Widmann2019] along with de facto standard, but biased ECE. The SKCE is an unbiased estimate of calibration. There might be some pitfalls of this metric that I'm not aware of, but the paper looks solid and convincing. Also, please put attention to Figure 83 in the arХiv version. Yes, ECE is the standard in the field, but it is the wrong standard that prevents us from meaningful scientific progress, so we should stop using it.

6) The results provided in Table 1 seem to be close values (0.6119 vs 0.6129, etc.), so at least standard deviations need to be reported. Also, there is no mentioning of several runs per results in the text.

The paper toches the nice topics but, overall it feels like "ok, but not enough". The theory is interesting but it does not give us a lot of insides (maybe it's very subjective). The dynamic temperature scaling is not proofed to outperform the basslines. The contributions feel disconnected. The writing quality needs to be improved.

Comments:

1) As far as I can tell, the citation "The weights assigned to the probabilities are either optimized using AUC as in (Ashukha et al., 2020) ..." is incorrect, as there is no mentioning of optimizing weights using AUC in the paper.

2) typo: It should be noted that for A finite number of sampleS

[Lakshminarayanan2017] Lakshminarayanan B, Pritzel A, Blundell C. Simple and scalable predictive uncertainty estimation using deep ensembles. In Advances in neural information processing systems 2017 (pp. 6402-6413).

[Ashukha2020] Ashukha A, Lyzhov A, Molchanov D, Vetrov D. Pitfalls of in-domain uncertainty estimation and ensembling in deep learning. ICLR, 2020.

[Vaicenavicius2019] Juozas Vaicenavicius, David Widmann, Carl Andersson, Fredrik Lindsten, Jacob Roll, and Thomas B Schon. Evaluating model calibration in classification. AISTATS, 2019.

[Widmann2019] Widmann D, Lindsten F, Zachariah D. Calibration tests in multi-class classification: A unifying framework. In Advances in Neural Information Processing Systems 2019 (pp. 12257-12267). https://arxiv.org/pdf/1910.11385.pdf

---

> ### Author Response · Authors · 2020-11-16
> **Response to AnonReviewer2**
>
> Q1:	The main question of the paper "Should ensemble members be calibrated?" feels trivial, because the community is aware of the simple example that provides an answer. The Deep Ensembles [Lakshminarayanan2017] have miscalibrated members---conventional DNNs, but the predictions of an ensemble are, in-most-cases, calibrated. Thus the answer is "No".
>
> Response:
> We think the question is non-trivial as ensembles are a standard approach for both improving the performance of deep learning systems, and deriving confidence measures associated with these predictions. Well calibrated systems are important for bot appropriate system combination and uncertainty.  The observation of mis-calibrated members lead to better calibrated ensemble doesn’t answer the question of “what if the members are calibrated”, does it lead to better calibrated ensemble? How should we treat the members, calibrate them or make them worse calibrated, in order to achieve better calibrated ensemble?  This paper attempts to address these questions.
> Second, the deep ensemble predictions are not always well calibrated. This paper confirms this statement both empirically and theoretically.
> Third, the empirical observation of “calibrating members leads to ensemble calibration degradation” has been reported in several previous papers, however, to the authors knowledge this is the first time a theoretical analysis of this observation has been given.
> Fourth, we do not only focus on this observation, but also discuss the calibration of non-prediction probabilities.  The analysis leads to the answer that is not simply “No”. Just to clarify this statement. The answer is “No,” when considering top-label calibration. The answer is “Yes”,if we are talking about all-label global calibration, when it is safe to say that “calibrated members lead to calibrated ensemble” (Proposition 2). We will try to be clearer in the text.
>
> Q2:	The paper mostly is clearly written, but section 4.1 "Theoretical Analysis" is extremely hard to follow.
>
> Response:
> We will update the text to make it clear.

---

> > ### Comment · AnonReviewer2 · 2020-11-16
> > **Q1 there are non-trivial result related to global calibration, but I'm still not seeing much potential**
> >
> > I appreciate the very swift response of the authors.
> >
> > Q1:
> >
> > > Response: ...  “what if the members are calibrated”, does it lead to better calibrated ensemble? How should we treat the members, calibrate them or make them worse calibrated, in order to achieve better calibrated ensemble?
> >
> > On the one hand, a fair point, on the other hand, an averaging of predictions generally increases the entropy of the mean prediction, so calibrated members will lead to more underconfident predictions as uncalibrated don't.
> >
> > > ... the deep ensemble predictions are not always well calibrated. This paper confirms this statement both empirically and theoretically.
> >
> > a. Do you mean the case of "all-label calibration"? It is interesting, but exact "all-label calibrated" seems to be a very unrealistic scenario for any non-trivial regions. Correct me if I'm wrong.
> >
> > b. Empirical results are tied on Q5 and Q6.
> >
> > While I'm more convinced now (either way, "triviality" is subjective property), but I, unfortunately, do not think that results are strong enough. I spend a decent amount of time thinking about how researchers can benefit from these results and did not find much. I'm saying it to highlight what prevents me to immediately increase my score. I'm open to other opinions of course.

---

> ### Author Response · Authors · 2020-11-16
> **Response to AnonReviewer2**
>
> Q3:	The calibration of ensemble has been proposed in [Ashukha2020, 5 Discussion & Conclusion]. ("The resulting ensemble predictions ..., requiring calibration functions to be optimized for the ensemble prediction, rather than ensemble members.")
>
> Response:
> We didn’t find the complete sentence the reviewer points us to in Section 5 Discussion & Conclusion.  The most relevant sentence is
> “Temperature scaling is a must even for ensembles. While ensembles generally have better calibration out-of-the-box, they are not calibrated perfectly and can benefit from the procedure.”
> The statement of “While ensembles generally have better calibration out-of-the-box” as shown in the empirical results does depend on the ensemble members, whch has also been reported in other  papers such as [Rahul Rahaman and Alexandre H Thiery. Uncertainty quantification and deep ensembles. 2020].
>
> The analysis in the paper is different to that in [Ashukha2020] in the following ways:
> First, [Ashukha 2020] doesn’t discuss in detail the relation between the calibration of members and that of the whole ensemble.
> Second, [Ashukha 2020] proposes the calibration method based on data augmentation at test time, which is different and complementary to our method of calibration at post-hoc training time. These two methods can actually be applied in a cascaded pipeline.
>
> Q4:	The two main contributions (4.1 Theoretical Analysis, 4.2 Temperature Annealing for Ensemble Calibration) feels not related, they are basically two independent topics packed in the one paper.
>
> Response:
> The relation is that after we have the theoretical analysis and know that we shouldn’t calibrate the member if we want to achieve better accuracy in Section 4.1, we then try to apply and improve temperature annealing to calibrate the ensemble as a whole in Section 4.2.  They are actually quite natural developments about the problem of ensemble calibration.
>
> Q5:	The empirical comparison exploits the calibrations score (e.g., ECE). ECE is a biased estimate of true calibration with a different bias for each model, so it is not a valid metric to compare different models (see Vaicenavicius2019). ...
> What I suggest is to use the squared kernel calibration error (SKCE) proposed in [Widmann2019] along with de facto standard, but biased ECE. ... so we should stop using it.
>
> Response:
> Yes, we agree that ECE is a biased metric and a better metric should be considered.  We focused on ECE as this is the standard metric used. As suggested by the reviewer we will give the performance use SKCE. However this does not alter the conclusions and nature of the analysis derived in the paper.
>
> Q6: The results provided in Table 1 seem to be close values (0.6119 vs 0.6129, etc.), so at least standard deviations need to be reported. Also, there is no mentioning of several runs per results in the text.
>
> Response:
> Yes, we only have one run.  We will run multiple experiments to show the deviations.
>
> Q7:	As far as I can tell, the citation "The weights assigned to the probabilities are either optimized using AUC as in (Ashukha et al., 2020) ..." is incorrect, as there is no mentioning of optimizing weights using AUC in the paper.
>
> Response:
> Yes, we originally intend to emphasize that AUC is not comparable between different models as discussed in [Ashukha et al., 2020] and thus not a good metrics for determining the combination weights.  We will add this back.

---

> > ### Comment · AnonReviewer2 · 2020-11-16
> > **Q3 Ok; Q4 Not Ok; Q5, Q6 corrections are needed;**
> >
> > Q3:
> >
> > > We didn’t find the complete sentence the reviewer points us to in Section 5 Discussion & Conclusion.
> >
> > I think it is a misunderstanding, as the sentence was brought from your paper. Specifically, it is taken from the conclusion of the submitted version. Thank you for commenting on the concern. I was just bringing to your attention that ensembles have been calibrated before (final predictions of an ensemble and not just separate members). It is worth properly cite the related works.
> >
> > Q4:
> >
> > > Response: The relation is that after we have the theoretical analysis and know that we shouldn’t calibrate the member if we want to achieve better accuracy in Section 4.1, we then try to apply and improve temperature annealing to calibrate the ensemble as a whole in Section 4.2. They are actually quite natural developments about the problem of ensemble calibration.
> >
> > I respectfully disagree with the response. As I see, the dynamic calibration has no relation to the other material proposed in the paper. What I'm trying to say is that paper should not be a collection of disconnected facts or methods, or at least it should be clearly indicated. If I'm wrong, please explain a) why do we need a new calibration method in the paper and b) how is it connected to the other material.
> > To be clear, that can not be a reason to reject a paper, but IMO this is a weak point.
> >
> > Q5, Q6:
> >
> > > Response: As suggested by the reviewer we will give the performance use SKCE.
> > > Response: Yes, we only have one run. We will run multiple experiments to show the deviations.
> >
> > I expect the corrections to be done due Nov 23, so we can discuss the final results.
> >
> > > However this does not alter the conclusions and nature of the analysis derived in the paper.
> >
> > Yes, but it will affect all the practical results.

---

> > > ### Author Response · Authors · 2020-11-23
> > > **Results Updated**
> > >
> > > We have updated the experimental results in Table 1, with deviations from 10 runs.  The SKCE is also added as calibration error metrics.
> > >
> > > Here are the explanation for the two questions:
> > > a) why do we need a new calibration method in the paper
> > > As revealed by the analysis based on regions, for different regions the calibration mapping functions from original predicted probability to the calibrated probability are possibly different.  Hence, we propose to learn the region-specific mapping functions of temperature scaling, this yield the dynamic temperature scaling as shown in this paper.
> > >
> > > b) how is it connected to the other material
> > > Actually, the conventional temperature scaling method using one temperature for all regions is a special case of our region-specific temperature scaling by using the same temperature for all regions.
> > >
> > > Also, the idea of dividing input space into regions for calibration is also investigated in previous literatures, like (Kuleshov & Liang 2015) as mentioned in the section of related work.  Our contribution is to investigate region-specific temperature scaling, while in  (Kuleshov & Liang 2015) the region-specific mapping functions are K-NN and decision trees.
> > >
> > > Volodymyr Kuleshov and Percy S Liang. Calibrated structured prediction. In Advances in Neural Information Processing Systems, pp. 3474–3482, 2015.

---

### Official Review · AnonReviewer4 · 2020-10-30
**Interesting observations, writing can be improved**

**Rating:** 6
**Confidence:** 3

**Review:**

Update after the author response: I've read the other reviews, and agree with R2 and R3. I think the paper is useful (emphasizes you need to calibrate the final ensemble, not enough to calibrate members), and has some nice conceptual contributions (explaining that if ensemble accuracy > average member accuracy (which is usually the case), and the ensemble is calibrated even in just a global/weak sense, then the members must be uncalibrated). This could spur more research into conceptually analyzing ensembles, and seems interesting. But I understand the other reviewer's concerns that it's not clear what practical impact this will have, so I'm keeping my score at a 6 (instead of raising to a 7).

#########################################################################

Summary:

This paper tackles the problem of calibrating an ensemble. They show experimentally that calibrating all members of an ensemble is often not enough to calibrate the combined ensemble, so instead we need to calibrate the final predictions of the ensemble. Additionally, they show that using a different temperature parameter for different regions of outputs can improve calibration. They explain why if the ensemble members are top-label calibrated (even in a very weak sense they call “global” calibration”), and the ensemble is calibrated, then the ensemble is less accurate than the average member of the ensemble.

#########################################################################

Reasons for score:

They make interesting observations about calibration of ensembles that could guide practitioners. For example, that it’s not enough to calibrate the members of the ensemble. They also raise an intriguing connection between calibration of ensemble members and ensemble accuracy, one would not expect a priori that if both are calibrated the ensemble would do worse than the average member. I could see this result being interesting to people who study ensembles as well. There are some weaknesses in writing and execution, but overall this paper is probably worth publishing if edited.

#########################################################################

Pros:

- I think it’s a nice observation that calibrating the members of an ensemble may not yield a calibrated ensemble. It’s easy to come up with toy examples where this is the case, but it’s interesting that this seems to be the case in practice.

- They make an intriguing observation that if the ensemble members are in fact calibrated and the ensemble is calibrated, then the ensemble accuracy is at most the average member accuracy


#########################################################################

Cons:

- I believe the writing can be substantially simplified. The core ideas are simple and nice, but it takes a lot of effort to get to them, and I believe the authors should put in more work into making this understandable.

- Some of the results seem unrealistic and can be omitted. For example in the start of section 4.1, the first couple of results require that the ensemble member regions and ensemble regions are the same. This seems rather unrealistic. The assumptions in prop 1 seem too strong to me. I’d remove the mentions of regions and I’d instead mention the other results (prop 2, 3, 4) in the main paper, Section 4.1. You could just move the propositions, and give some intuition for why the results are true. Removing regions should also considerably simplify the notation and setup.

- I’m not quite sure what you mean in the intro when you say “Eq. (1) doesn’t explicitly reflect the relation between … and the underlying data distribution p(x, y)”. The definition in Equation (1) uses p(x, y). I’m not sure why all the definitions in 3.1 and 3.3 are defined in a way different from the standard ways in the calibration literature e.g. in Kull et al 2019 or Kumar et al 2019.

- Temperature scaling is performed on logits, not on the actual probabilities. From equations 24, 25, and 26 it looks like you might be doing temperature scaling on the probability space (in equation 24, 25 the first argument to f is the probability, not the logit), which looks a bit odd.

- Prop 4 should also hold when K = 2 (2 ensemble members) I believe. Happy to provide an example.

- Some symbols are undefined. For example, \delta(y^{(i)}, \omega_j), I don’t believe \delta is defined. I think it should be 1 if they are equal and 0 otherwise?

#########################################################################

Questions and things to improve:

- Please answer the cons above.

- Ensembles are particularly useful because they tend to be more calibrated out of domain (Lakshminarayanan et al 2017). It could be useful to see which of these methods (calibrating the members, or the entire ensemble) is better calibrated when we have domain shift (e.g. training data = CIFAR-10, test data = CIFAR-10C, Hendrycks et al 2019).

- Having confidence intervals for the calibration errors would be nice (and also using more modern, debiased estimators to estimate the calibration error) e.g. in Kumar et al 2019.

#########################################################################

All cites mentioned are already in the paper, except:
Benchmarking Neural Network Robustness to Common Corruptions and Perturbations. Dan Hendrycks, Thomas Dietterich. ICLR 2019.

---

> ### Author Response · Authors · 2020-11-16
> **Response to AnonReviewer4**
>
> Q1:
> •	I believe the writing can be substantially simplified. The core ideas are simple and nice, but it takes a lot of effort to get to them, and I believe the authors should put in more work into making this understandable.
> •	Some of the results seem unrealistic and can be omitted. For example in the start of section 4.1, the first couple of results require that the ensemble member regions and ensemble regions are the same. This seems rather unrealistic. The assumptions in prop 1 seem too strong to me. I’d remove the mentions of regions and I’d instead mention the other results (prop 2, 3, 4) in the main paper, Section 4.1. You could just move the propositions, and give some intuition for why the results are true. Removing regions should also considerably simplify the notation and setup.
>
> Response:
> Thanks for the comments, we will update the text to make it clear.
> =====================
>
> Q2: I’m not quite sure what you mean in the intro when you say “Eq. (1) doesn’t explicitly reflect the relation between … and the underlying data distribution p(x, y)”. The definition in Equation (1) uses p(x, y). I’m not sure why all the definitions in 3.1 and 3.3 are defined in a way different from the standard ways in the calibration literature e.g. in Kull et al 2019 or Kumar et al 2019.
>
> Response:
> As shown by equation (14), our definition of ECE is equivalent to ECE defined in Guo et al. 2017 and TCE in Kumar et al. 2019.  Actually, the definitions in Guo et al. 2017 and Kumar et al. 2019 are using a sample-based definition.  However, we are defining the ECE based on the true distribution p(x,y), where the samples are drawn.  We adopted this definition, based on the true distribution, as it allows more fundamental definitions of the attributes of calibration and ensemble calibrstion.  We also show in Appendix A.3 Example 2 that with finite number of samples, the calibration error calculated based on the sample-based definition can be different from the calibration error obtained via the true distribution.
> ======================
>
> Q3: Temperature scaling is performed on logits, not on the actual probabilities. From equations 24, 25, and 26 it looks like you might be doing temperature scaling on the probability space (in equation 24, 25 the first argument to f is the probability, not the logit), which looks a bit odd.
>
> Response:
> Thanks for the comment, the function f in Eq. (24) and (25) is actually referring to a general mapping from one probability to another probability.  The function f in Eq. (26) is a special case of the general mapping.  We have updated the text to make it clear.
> ======================
>
> Q4:	Prop 4 should also hold when K = 2 (2 ensemble members) I believe. Happy to provide an example.
>
> Response:
> No, when K=2, if all members are globally top-label calibrated, then the ensemble is 'always' global top-label calibrated.  Because when K=2, top-label calibration is equivalent to all-label calibration.  According to proposition 2, we know the above holds.
> ======================
>
> Q5:	Some symbols are undefined. For example, \delta(y^{(i)}, \omega_j), I don’t believe \delta is defined. I think it should be 1 if they are equal and 0 otherwise?
> Response:Thanks for the comment, we have added the definition for \delta function.
> ======================
>
> Q6:
> •	Ensembles are particularly useful because they tend to be more calibrated out of domain (Lakshminarayanan et al 2017). It could be useful to see which of these methods (calibrating the members, or the entire ensemble) is better calibrated when we have domain shift (e.g. training data = CIFAR-10, test data = CIFAR-10C, Hendrycks et al 2019).
> •	Having confidence intervals for the calibration errors would be nice (and also using more modern, debiased estimators to estimate the calibration error) e.g. in Kumar et al 2019.
> All cites mentioned are already in the paper, except: Benchmarking Neural Network Robustness to Common Corruptions and Perturbations. Dan Hendrycks, Thomas Dietterich. ICLR 2019.
>
> Response: Thanks for these constructive comments.  We will improve our experiments accordingly and add the reference.

---

> > ### Comment · AnonReviewer4 · 2020-11-23
> > **Thanks for clarifications, interesting observations with conceptual explanations, so weak accept**
> >
> > Thank you for clarifying these details.
> >
> > Kumar et al. 2019 do not use a sample based definition, they give a population definition based on p(x, y) in equations 1 and 2.
> >
> > I have read the other reviews, and would still give the paper a weak accept. I think the central observation is interesting: that if ensemble accuracy > average member accuracy (which is usually the case), and the ensemble is calibrated even in just a global/weak sense, then the members must be uncalibrated. They give a nice conceptual explanation. I agree with R1 that the explanation is not very complicated, but that seems like a strength.
> >
> > R1 makes a good point about the other papers showing this experimentally (without a conceptual explanation as far as I can tell), but they only came about 2 months before the ICLR deadline, on Arxiv, and don't have a conceptual explanation (I think?) - so it doesn't seem like a big deal.
> >
> > The notation still seems unnecessarily complicated, and it seems like the whole discussion about regions can be moved to the Appendix - I'm not seeing anything conceptually interesting in the region discussion.

---

> > ### Comment · AnonReviewer4 · 2020-11-23
> > **Clarify that global all label calibration does not imply global top label calibration**
> >
> > Below prop 2, I think it's worth clarifying that global all label calibration does not imply global top label calibration, if you haven't already. Global all label calibration sounds strictly stronger, but it isn't (the authors definitely seem to know this, but it's nice to clarify this for the reader).

---

> > ### Comment · AnonReviewer4 · 2020-11-23
> > **Confidence interval calculation**
> >
> > Just for clarity, can you clarify how you calculated the confidence intervals for the ECE in the Appendix?

---

### Official Review · AnonReviewer3 · 2020-10-30
**Theoretically backed explanations for expected conclusions**

**Rating:** 6
**Confidence:** 3

**Review:**

The paper makes an analysis of calibration in ensembles of deep learning models. Through some theoretical developments, the paper supports that a given ensemble cannot be more confident than the average individual members for regions where the ensemble is well calibrated. Empirical results, on CIFAR-100 and three different deep models, report a comparison of ensemble calibration, where calibration is done over all members in order to achieved a calibrated ensemble decision, over individual calibration of members with no feedback from the ensemble decisions. Results show that individual member calibration does not lead to calibrated ensembles, and as such calibrating directly on the ensemble output is required for obtained a proper evaluation of its uncertainty. Different ensemble calibration approaches are also compared.

Pros:
- Overall well-written paper.
- Straightforward proposal, simple yet meaningful on several aspects for better understanding of the link between calibration and ensembles.
- Rely on theory to support some claims, which strengthen the proposal.

Cons:
- The proposal is somewhat trivial, although I do not have knowledge that it has been investigated in detail elsewhere. Before reading the paper, I expected the results (i.e. calibration of individual members will not lead to calibrated ensemble decisions; calibration at the ensemble level is required), the paper is somewhat confirming this in a more explicit manner.
- Evaluation on only one dataset (CIFAR-100) in the main paper, with another dataset for the appendix (CIFAR-10).
- Results on CIFAR-10 in the appendix are not very compelling.
- It is hard to make sense of the results in Table 1 and similar. Differences are small and difficult to interpret.
- The explanations and organization of the paper are hard to following in some specific part.

Although the paper is making a well-founded analysis of a hot topic in the last few years (i.e.,  ensembles are a way to evaluate uncertainty on decisions), I found it having some relatively trivial developments. And the conclusion is intuitive and expected. However, it is the first time I see this point well articulated, and the authors have made a good effort to develop theoretically backed explanations to support this.

---

> ### Author Response · Authors · 2020-11-16
> **Response to AnonReviewer3**
>
> Response:The empirical observation of “ensemble of calibrated or uncalibrated members leads to calibration degradation” has been reported in several previous papers, however, to the authors knowledge this is the first time that a theoretical analysis of this observation has been given. Note we do not only focus on this observation, where only the predictions from the members and ensemble are considered.  The analysis also leads to interesting conclusions about all class calibration.
>
> Thank the reviewer for the valuable comments!

---

### Official Review · AnonReviewer1 · 2020-11-01
**AnonReviewer1**

**Rating:** 4
**Confidence:** 3

**Review:**

- **Summary**:
This paper investigates the impact of different calibration strategy (pre-combination, post-combination and its dynamic variant) on the performance of a deep ensemble. It presents both theoretical and empirical proof to show that well-calibrated ensemble member does guarantee calibration in the final ensemble.

- **Strength**:
  * A coherent theoretical account for the issue of calibrating Deep ensembles. Accompanied by empirical evidence from CIFAR datasets.
  * Although not stated explicitly, a new calibration approach (dynamic calibration) is introduced, which empirically leads to better performance.

- **Weakness**
  * Novelty may be limited: one central contribution of this paper is to provide a mathematical derivation to confirm the observation made in Rahaman and Thiery (2020) and Wen et al. (2020).  Although I appreciate author's work on providing mathematical explanation for recent empirical findings, I'm not sure if the submission in its current form  is contributing significant novel theoretical insight beyond the fact that ensemble prediction is less confidence, since max of the mean probability is no greater than mean of the max probabilities. On the other hand, the empirical investigation is conducted on a single vision task (CIFAR-10/-100). This paper can be made stronger by investigating synthetic situation where the ground truth is known, or extend experiment to also other data modalities (like  Guo et al. (2017)).

  * Organization: Given the place of the new approach (dynamic temperature scaling) in the experiment, it might be worthwhile to devote some paragraph to introduce the procedure in more detail.

- **Recommendation**: Based on reason stated in weakness, I recommend rejection since the either theoretical or the empirical contribution of this paper does not seem to be substantive enough for ICLR.

---

> ### Author Response · Authors · 2020-11-16
> **Response to AnonReviewer1**
>
> Q1: Novelty may be limited: one central contribution of this paper is to provide a mathematical derivation to confirm the observation made in Rahaman and Thiery (2020) and Wen et al. (2020). Although I appreciate author's work on providing mathematical explanation for recent empirical findings, I'm not sure if the submission in its current form is contributing significant novel theoretical insight beyond the fact that ensemble prediction is less confidence, since max of the mean probability is no greater than mean of the max probabilities. On the other hand, the empirical investigation is conducted on a single vision task (CIFAR-10/-100). This paper can be made stronger by investigating synthetic situation where the ground truth is known, or extend experiment to also other data modalities (like Guo et al. (2017)).
>
> Response:
> (1)	 In this work we are not only talking about the ensemble prediction, but examining the attributes of two type of calibration performance metrics: top-label (prediction) calibration; and all-label (prediction and non-prediction) calibration. We show that for top-label global calibration, the calibrated ensemble will be less accurate than the mean accuracies of calibrated members, which implies that we should leave the members uncalibrated in order to achieve ensemble accuracy that is better than the mean accuracy of members.  We also show that for all-label global calibration, then it is possible to say that “calibrated members lead to global calibrated ensemble” (Proposition 2).
> (2)	“This paper can be made stronger by investigating synthetic situation where the ground truth is known”—We actually have the experiments on synthetic dataset where ground truth confidence is known in the Appendix A.3 Example 2.
> (3)	Extended to other modalities.  Thanks for the comment.  Yes, it is desirable to verify our performance of dynamic temperature calibration on more modalities.   We will finish this in our future work.  The experiments are undergoing.
> =======================
>
> Q2: Organization: Given the place of the new approach (dynamic temperature scaling) in the experiment, it might be worthwhile to devote some paragraph to introduce the procedure in more detail.
>
> Response:
> We have the equation for dynamic temperature scaling (26). But due to the lack of space, we didn’t reveal much detail about dynamic temperature scaling.  We have added the discussion of region numbers in Section 4.3 Figure 3.
> =======================
>
> Thank the reviewer for the valuable comments!

---

### Public Comment · ~Jize_Zhang1 · 2020-11-16
**Related work on ensemble + calibration**

Great work on calibrating ensembles! I also want to refer to our recent work on improving post-hoc calibration methods using ensembles [1]. In addition, we also provided a binning-free KDE-based estimator to reduce the bias and binning sensitivity issues of existing histogram ECE estimators. All codes are also available online.

[1] Jize Zhang, Bhavya Kailkhura, and T Han. "Mix-n-Match: Ensemble and compositional methods for uncertainty calibration in deep learning.", ICML 2020, https://arxiv.org/pdf/2003.07329.pdf

---

> ### Author Response · Authors · 2020-11-18
> **Thanks for pointing us to the related paper**
>
> Thanks for your comment!  The paper is very related to ours.  We will study and use the estimator in your paper later.  Thank you!

---

### Author Response · Authors · 2020-11-18
**Overall Response**

We thank the reviewers’ comments very much! The paper has been updated accordingly. Further experiments are undergoing.  We would like to clarify the concerns from the reviewers globally from the following aspects.

(1) Why is this paper interesting to the community?

    This paper investigates the question of “Should Ensemble Members be Calibrated?”, which is motivated by the observations that ensembling calibrated or uncalibrated members suffers from calibration degradation. Though these observations are reported or revealed in several previous publications, the deep reason is unknown. This paper examines this question and to the authors knowledge this is the first time the theoretical reasons are given for explaining the phenomenon. Through our analysis, it turns out that the answer to this question is not trivial, neither simply “yes” nor “no”.

    We found that accuracy is naturally connected to global top-label calibration. If we want the calibrated ensemble’s accuracy to be higher than mean accuracy of members, the members shouldn’t be global top-label calibrated. Because the ensemble is less or equal confident than any of its members on the top label. However, if the members are global all-label calibrated, then the ensemble is always global all-label calibrated, no matter the ensemble’s accuracy exceeds the mean accuracy of members or not, which implies we should calibrate members to obtain calibrated ensemble.

(2) Why do we need the discussion on regions in this paper?

    The above inconsistent conclusions for ensemble calibration then motivate us to look into more detail, and we found that this is related to the regions that members are calibrated on. As a starting point, in Proposition 1, if members are calibrated on the same regions, the conclusions for top-label and all-label calibrations are consistent. Proposition 2 and 3 follow directly from Proposition 1 to discuss the scenario of minimum requirement of calibration, i.e. the global calibration. For all-label calibration, since the regions for all members are the same, i.e. the whole space of the input x, the calibrated members always lead to calibrated ensemble. Actually, the ensemble is still calibrated on the same region of whole space of x. However, this is not the case for top-label calibration, different members are not always calibrated on the same regions. Even when the members are top-label calibrated on the same regions, the regions for the resulting ensemble can still be different. This explains the inconsistency of conclusions for all-label and top-label calibration.

    However, when the regions are neither as trivial as Proposition 1 nor as the global minimum requirement, we are not able to generate any theoretical results.  But the regions give the intuitive explanation. This is worth the attention from the community.

(3) Are the different parts of this paper closely related?

    After we have the theoretical analysis and know that we shouldn’t calibrate the member if we want to achieve better accuracy in Section 4.1, we then try to apply and improve temperature annealing to calibrate the ensemble as a whole in Section 4.2.  They are actually quite natural developments about the problem of ensemble calibration. As shown in Equation (2), different regions require different degrees of confidence adjustment, hence, it is natural to change the global temperature to region-specific temperatures.  Experiments also shows the advantage of region-specific temperatures. The previous discussion focuses on the uniform combination weights for ensemble, motivated from a Bayesian approximation perspective. When different weights are desired, for example, in the scenario of different structures are combined to an ensemble, whether the above conclusions still hold.  In summary, the different parts of the paper are closely related to the logic of calibrating ensembles based on regions.

---

> ### Comment · AnonReviewer2 · 2020-11-24
> **Squared kernel calibration error (SKCE) results**
>
> I did a quick look at SKCE results (that is an unbiased metric of calibration, I was asking in my review), it seems that according to it dynamic scaling does not help much. Also in Table 5, the wrong number is highlighted with bold 447.50±1.78 < 568.94±7.07. Also, the results presented in Table 5 are different orders of magnitude compared to Fig. 83 of the original work https://arxiv.org/pdf/1910.11385.pdf. That causes some worries.
>
> I would say, it feels like it might be an error in implementation --- then we can tell nothing, or a metric strongly disagreement with ECE in this particular case.

---

> > ### Author Response · Authors · 2020-11-24
> > **SKCE results**
> >
> > Thank for your comments!
> >
> > There are at least two differences between the results in the Fig. 83 and our Table 5:
> >
> > (1) In Table 5, those are ensemble, not single models. While in Fig. 83, they are single models.
> > (2) We use LeNet 5, DenseNet 100, DenseNet 121 and 'Wide' ResNet 28 in Table 5.  However, we can only find DenseNet 121 in Fig. 83.  Those ResNets are not wide ResNets.
> >
> > Oder difference.
> >
> > (1) Even in Fig. 83, the orders of unbiased linear SKCE (SKCE_ul) and the unbiased quadratic SKCE (SKCE_uq) are different.  This is an interesting observation we can look at in the next step.
> > (2) Different orders between SKCE and ECE are also observed in Fig. 83.
> >
> > Why ECE observes slight improvement while SKCE not?
> >
> > One possible explanation is that we optimize the temperatures with respect to ECE, as stated in our paper.  It is interesting to investigate optimizing the temperatures with respect to SKCE as well.

---

> > > ### Comment · AnonReviewer2 · 2020-11-24
> > > **SKCE results**
> > >
> > > > One possible explanation is that we optimize the temperatures with respect to ECE, as stated in our paper. It is interesting to investigate optimizing the temperatures with respect to SKCE as well.
> > >
> > > Yes, exactly. Or maybe choose some outer criteria like NLL?

---

> > > > ### Author Response · Authors · 2020-11-24
> > > > **NLL criterion**
> > > >
> > > > Thank you for your swift response!
> > > >
> > > > Yes, we agree NLL is one of the criteria that we can use, as also provided in Table 5.
> > > >
> > > > NLL actually supports our proposal better than ECE that dynamic temperature is better.  But we don't want to stop there, because NLL depends on both accuracy and calibration, and the comparison of NLL is effective for revealing calibration performance only when the accuracies of compared models are the same.  This concern is also illustrated in (Ashukha et al. 2020).  Therefore we only use NLL as an auxiliary criterion, but rely more on ECE or SKCE.
> > > >
> > > > Arsenii Ashukha, Alexander Lyzhov, Dmitry Molchanov, and Dmitry Vetrov. Pitfalls of in-domain uncertainty estimation and ensembling in deep learning, ICLR 2020

---

### Decision · Program_Chairs · 2021-01-07
**Final Decision**

**Decision:**

Reject

**Comment:**

This paper studies ensemble calibration and the relationship between the calibration of individual ensemble member models with the calibration of the resulting ensemble prediction.  The main theoretical result is that individual ensemble members should not be individually calibrated in order to have a well-calibrated ensemble prediction.  While other recent work has found this to be the case in empirical results, this paper substantiates the empirical results through theoretical results.

Pros:
* Theoretical study of ensemble calibration with meaningful insights

Cons:
* Contributions limited to theoretical study of known observation and dynamic temperature scaling.
* Dynamic temperature scaling is not shown to outperform baseline methods.
* Limited experimental validation: CIFAR-10/CIFAR-100.

The authors engaged in a extensive discussion with reviewers and made changes to their paper, including adding standard deviation results over multiple runs and the SKCE calibration measure.

Overall this is solid work and could be accepted to the conference; however, reviewers agree that parts of the work are lacking, in particular: 1. limited experimental evaluation (one type of task, one/two datasets only), and 2. given known literature the benefit of the derived theoretical results to practioners is not clear.  The discussions have been unable to resolve this disagreement.